# Structure-based development and preclinical evaluation of the SARS-CoV-2 3C-like protease inhibitor simnotrelvir

Xiangrui Jiang[1,2,11], Haixia Su[1,11], Weijuan Shang[3,11], Feng Zhou [4,5,11], Yan Zhang[1,11], Wenfeng Zhao[1,11], Qiumeng Zhang[1], Hang Xie[1,6], Lei Jiang[5], Tianqing Nie[1,6], Feipu Yang[1], Muya Xiong[1,2], Xiaoxing Huang[5], Minjun Li[7], Ping Chen[8], Shaoping Peng[4,8], Gengfu Xiao [3], Hualiang Jiang [1,12], Renhong Tang[4,5] ✉, Leike Zhang [3,9] ✉, Jingshan Shen [1,2] ✉ & Yechun Xu [1,2,10] ✉

The persistent pandemic of coronavirus disease 2019 (COVID-19) caused by severe acute respiratory syndrome coronavirus 2 (SARS-CoV-2) and its variants accentuates the great demand for developing effective therapeutic agents. Here, we report the development of an orally bioavailable SARS-CoV-2 3C-like protease (3CL^pro) inhibitor, namely simnotrelvir, and its preclinical evaluation, which lay the foundation for clinical trials studies as well as the conditional approval of simnotrelvir in combination with ritonavir for the treatment of COVID-19. The structure-based optimization of boceprevir, an approved HCV protease inhibitor, leads to identification of simnotrelvir that covalently inhibits SARS-CoV-2 3CL^pro with an enthalpy-driven thermodynamic binding signature. Multiple enzymatic assays reveal that simnotrelvir is a potent pan-CoV 3CL^pro inhibitor but has high selectivity. It effectively blocks replications of SARS-CoV-2 variants in cell-based assays and exhibits good pharmacokinetic and safety profiles in male and female rats and monkeys, leading to robust oral efficacy in a male mouse model of SARS-CoV-2 Delta infection in which it not only significantly reduces lung viral loads but also eliminates the virus from brains. The discovery of simnotrelvir thereby highlights the utility of structure-based development of marked protease inhibitors for providing a small molecule therapeutic effectively combatting human coronaviruses.

The catastrophic coronavirus disease 2019 (COVID-19) pandemic caused by severe acute respiratory syndrome coronavirus 2 (SARS-CoV-2) and its variants has resulted in an unbearable number of infections and deaths worldwide, and posed an unprecedented threat to global public health and economies. Thus, great efforts have been devoted to developing vaccines as well as antiviral agents for the treatment and prevention of COVID-19, but the clinically significant therapeutic options for COVID-19 are limited. According to therapeutic strategies against other viruses, such as HCV and HIV, orally effective small molecule drugs targeting viral proteins that are essential for virus replication have been particularly pursued to combat the ongoing pandemic and future SARS-like zoonotic coronavirus outbreaks.

A full list of affiliations appears at the end of the paper. ✉e-mail: renhong.tang@simceregroup.com; zhangleike@wh.iov.cn; shenjingshan@simm.ac.cn; ycxu@simm.ac.cn

Coronaviruses share key genomic elements that provide promising therapeutic targets, such as spike glycoprotein, RNA-dependent RNA polymerase (RdRp) and 3C-like protease (3CL[pro])[1]. 3CL[pro] and a papain-like protease (PL[pro]) are responsible for the cleavage of two viral polyproteins (pp1a and pp1ab) that result from translation of the viral genomic RNA by host ribosomes, yielding nonstructural proteins that are vital to viral replication[2]. There are 11 3CL[pro] cleavage sites in polyproteins[2], therefore, 3CL[pro] is also referred to as the main protease (M[pro]). 3CL[pro] has a unique ability to recognize substrates for specificity and cleaves at a position next to glutamine (Leu-Gln ↓ (Ser, Ala, Asn, Gly)), distinguishing coronavirus 3CL[pro] from related human proteases and thereby providing a natural 'safety valve' to minimize the adverse effects of 3CL[pro] inhibition[2]. Furthermore, unlike the spike glycoprotein, which is vulnerable to mutations[3], the structure of 3CL[pro] and particularly the substrate binding cavity are highly conserved across a range of pathogenic coronaviruses, such as SARS-CoV, Middle East respiratory syndrome coronavirus (MERS-CoV), and SARS-CoV-2 and its variants of concern (VOCs)[2]. Accordingly, 3CL[pro] is an attractive target for the development of broad-spectrum anti-CoV agents.

The inherent plasticity of the 3CL[pro] substrate binding cavity remains a major challenge in small molecule inhibitor design but lays a foundation for creating novel inhibitors with different chemical scaffolds. As a cysteine proteinase, substrate analogs or mimetics attached with a chemical warhead targeting the catalytic cysteine are generally designed as peptidomimetic inhibitors of 3CL[pro] with a covalent mechanism of action. Among them, nirmatrelvir (PF-07321332)[4], which was developed by Pfizer is an orally bioavailable inhibitor bearing a nitrile warhead and was the first compound to receive Emergency Use Authorization (EUA) by the US FDA for mild-to-moderate disease in patients with a minimum age of 12. Nirmatrelvir was commercialized under the brand name Paxlovid™ in combination with ritonavir, which acts as a pharmacokinetic (PK) booster. Very recently, the noncovalent, nonpeptidic, orally bioavailable small molecule SARS-CoV-2 3CL[pro] inhibitor ensitrelvir (S-217622) developed by Shionogi[5] has also been granted EUA in Japan for COVID-19 therapeutic intervention.

Given the excellent druggability of 3CL[pro] and the persistent demand to enrich our armory to fight against the growing number of drug-resistant SARS-CoV-2 variants or newly emergent pathogenic coronaviruses, there is continuing interest in the creation of novel 3CL[pro] inhibitors. We previously reported on the discovery of FB2001[6], a peptidomimetic covalent inhibitor harboring an aldehyde warhead, which is now in a phase II/III clinical trial (NCT05445934) as an intravenous treatment option for COVID-19. We describe here our effort toward the design of a newly approved drug, simnotrelvir (SSD8432 or SIM0417), which has been commercialized under the brand name XIANNUOXIN™ in combination with ritonavir for adult patients with mild-to-moderate COVID-19 in China. Guided by crystal structures and thermodynamic binding signature of SARS-CoV-2 3CL[pro] in complex with inhibitors, a hit-to-lead-to-candidate optimization was carried out to replace the warhead, P1, P2, and P4 segments of boceprevir in order to improve the potency against SARS-CoV-2 3CL[pro]. The resulting simnotrelvir utilizes a covalent mode of action to gain its potent antiviral activity both in enzymatic and cellular assays. Moreover, the preclinical studies including the in vitro off-target selectivity, in vivo pharmacokinetic properties, and in vivo safety profiles of simnotrelvir suggest that it is able to achieve an oral plasma concentration exceeding the in vitro antiviral cell potency in particular with co-administration of ritonavir. As expected, simnotrelvir in combination with ritonavir effectively blocks replications of the SARS-CoV-2 Delta variant in K18-hACE2 transgenic mice. Ultimately, with a well-characterized mechanism of action, effective antiviral activity in vitro and in vivo, oral bioavailability, and good safety, simnotrelvir serves as a clinical candidate for the treatment of COVID-19.

## Results

### Structure-based design of oral SARS-CoV-2 3CL[pro] inhibitors

To rapidly yield a covalent SARS-CoV-2 3CL[pro] inhibitor serving as an orally active clinical candidate to combat COVID-19, we initially carried out a structure-based antiviral drug discovery based on boceprevir. Boceprevir is an oral clinical drug that targets HCV NS3/4 A serine protease[7], and incorporation of its structural features may thereby improve the oral bioavailability of 3CL[pro] inhibitors. Boceprevir exhibits weak inhibition of SARS-CoV-2 3CL[pro] with a half-maximal inhibitory concentration (IC$_{50}$) of 8596 nM (Fig. 1a, Supplementary Fig. 1). The enzyme active site cavity of SARS-CoV-2 3CL[pro] contains a C145/H41 catalytic dyad, a canonical oxyanion hole formed by the main chain amide NHs of G143/S144/C145, and subsites S1' through S4 that are responsible for selectively recognizing either substrate residues or P1'-P4 segments of inhibitors. The binding mode of boceprevir in the X-ray crystal structure with SARS-CoV-2 3CL[pro] reveals that the electrophilic ketone warhead of this α-ketoamide inhibitor reacts with the C145 thiolate to produce a hemithioketal adduct while the P1'-amide, P1-cyclobutyl, P2-dimethyl cyclopropyl-proline, P3-tert-leucine, and P4-tert-butylamide segments of boceprevir individually bind to the S1' through S4 subsites, respectively[8] (Fig. 1b, Supplementary Fig. 2a). It has been clearly shown that the carbonyl oxygen of the P1'-amide forms two hydrogen bonds (H-bonds) with residues in the oxyanion hole, whereas the fully hydrophobic P1-cyclobutyl group only partially occupies the hydrophilic S1 subsite lined by the side chains of S144/H163/E166/H172 (Supplementary Fig. 2a). Due to the exclusive substrate preference of 3CL[pro] for P1-glutamine, a rigidified glutamine with a five-membered γ-lactam ring is widely used as the P1 segment in potent 3CL[pro]-selective inhibitors such as GC376[9], PF-07304814[10], FB2001[6], and nirmatrelvir[4] (Supplementary Fig. 3). In addition, it has been reported that a small electrophilic warhead, such as an aldehyde, is more potent than a bulky warhead in 3CL[pro] inhibitors[11–13]. Based on these findings, we created compounds **1**, **2**, and **3** bearing a small warhead composed of an aldehyde, a nitrile and an α,β-unsaturated ketone, respectively, and the P1-γ-lactam ring, which led to significantly improved SARS-CoV-2 3CL[pro] biochemical potency of the three derivatives compared to boceprevir (Fig. 1c). It is encouraging that the IC$_{50}$ values of these three compounds determined by a fluorescence resonance energy transfer (FRET)-based enzymatic assay were 9, 20, and 14 nM (Supplementary Fig. 1), respectively, yielding a > 420-fold improvement in biochemical potency relative to boceprevir.

Crystal structures of SARS-CoV-2 3CL[pro] in complex with compounds **1-3** were determined at 1.5-2.0 Å resolutions, and these compounds exhibited an essentially identical binding mode in the substrate-binding cavity of the protease (Fig. 2a, Supplementary Fig. 2b). A covalent bond formed between the aldehyde, nitrile or α,β-unsaturated ketone warhead and the catalytic C145, and as anticipated, the P1-γ-lactam ring displays a perfect geometric fit to engage the S1 subsite and forms productive H-bonds to the side chains of H163/E166 and the F140 main chain. Those favorable H-bonding interactions are missing in the interaction between boceprevir with SARS-CoV-2 3CL[pro] (Supplementary Fig. 2), partially accounting for its low inhibition against this protease. Nevertheless, in a way very similar to boceprevir, the carbonyl oxygen of the aldehyde and the α,β-unsaturated ketone or the nitrogen of the nitrile forms multiple H-bonds with the residues in the enzyme's oxyanion hole, and the P2-dimethyl cyclopropyl-proline, P3-tert-leucine and P4-tert-butylamide moieties of the three compounds bind to the S2, S3, and S4 subsites, respectively (Fig. 2a, Supplementary Fig. 2). Accordingly, replacement of the P1 segment of boceprevir with the P1-γ-lactam and aldehyde, nitrile or α,β-unsaturated ketone warhead affords an appreciable increase in the biochemical potency of the inhibitors.

Since a cysteine-targeted covalent ligand with proper warhead reactivity is critical to achieve prolonged efficacy and reduce promiscuity-induced side effects, the half-life of the electrophile

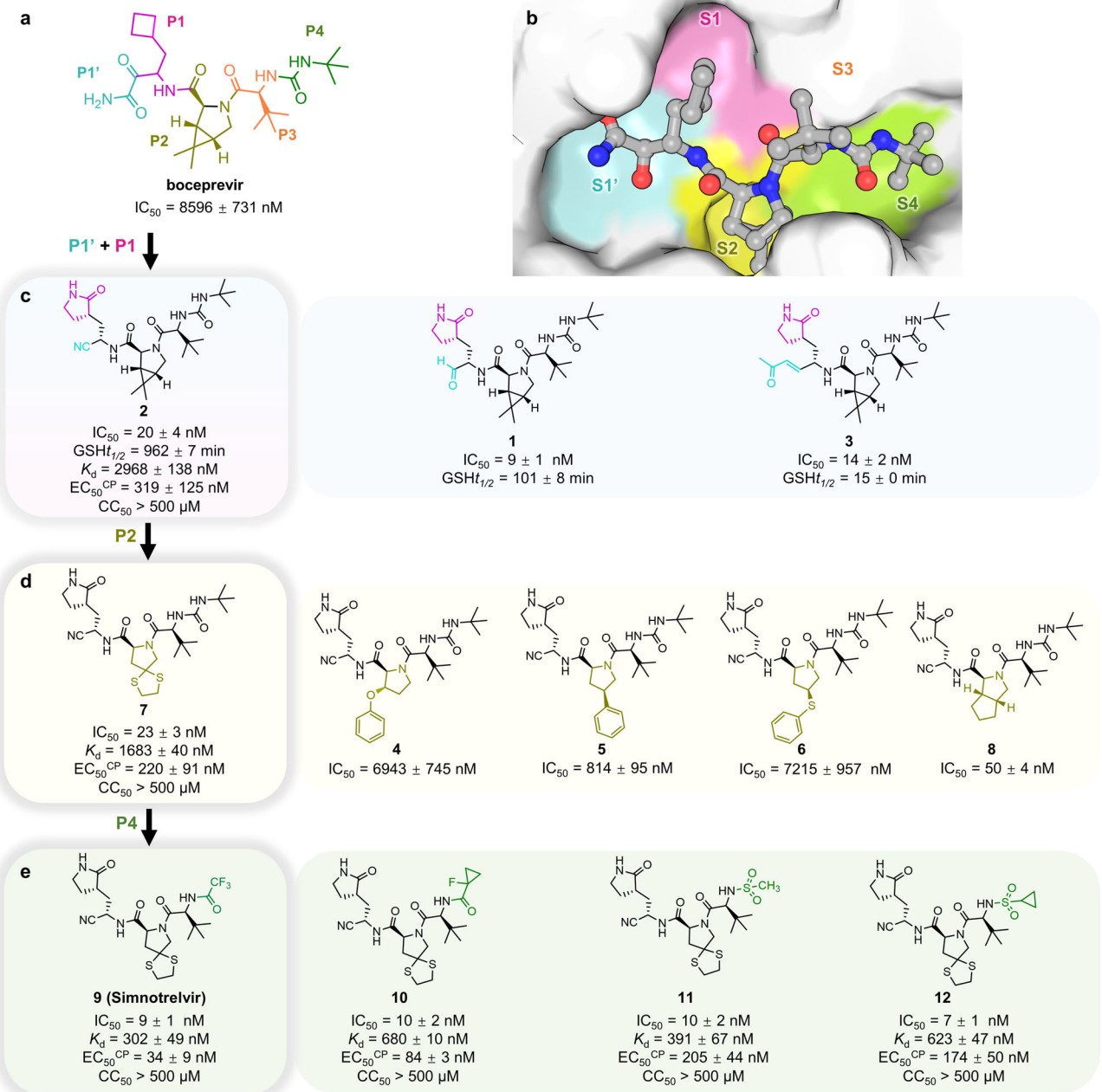

**Fig. 1 | Design strategy, chemical structures, and in vitro biochemical and antiviral activities of SARS-CoV-2 3CL$^{pro}$ inhibitors. a** The chemical structure of boceprevir and its inhibitory activity against SARS-CoV-2 3CL$^{pro}$ (expressed as IC$_{50}$). P1′-P4 segments of boceprevir are colored in cyan, magenta, olive, orange, and green, respectively. **b** Molecular surface representation of boceprevir interacting with the S1′-S4 subsites of SARS-CoV-2 3CL$^{pro}$ (PDB code: 6XQU). Boceprevir is shown in gray balls and sticks. The S1′-S4 subsites are colored cyan, magenta, yellow, orange, and green, respectively. **c**–**e** Chemical structures and in vitro biochemical and biophysical characterization of compounds that were designed to favorably interact with the S1′(C145) and S1 (**c**), S2 (**d**) and S4 (**e**) subsites of SARS-

CoV-2 3CL$^{pro}$. IC$_{50}$ is the half-maximal inhibitory concentration of the compound against the protease determined by the enzymatic inhibition assay. $K_d$ is the dissociation constant of the compound from the C145G mutant 3CL$^{pro}$ measured by ITC. GSH$t_{1/2}$ is the half-life of the electrophile warhead to react with glutathione. EC$_{50}$$^{CP}$ and CC$_{50}$ are the half-maximal effective concentration in the presence of the P-glycoprotein efflux inhibitor CP-100356 and the half-maximal cytotoxic concentration of the compound in Vero E6 cells, respectively. Three independent experiments were performed to determine the IC$_{50}$, GSH$t_{1/2}$, $K_d$, EC$_{50}$$^{CP}$, and CC$_{50}$ values.

warhead to react with glutathione (GSH$t_{1/2}$) was determined to evaluate the inherent reactivity of different warheads in the three inhibitors. The GSH$t_{1/2}$ values of compounds **1**, **2**, and **3** were 101, 962, and 15 min (Supplementary Fig. 4), respectively, indicating that the α,β-unsaturated ketone group in **3** has the highest electrophilicity, while the nitrile in **2** retains the lowest reactivity in this regard. In addition, the antiviral activity of compound **2** against the SARS-CoV-2 Delta strain in Vero E6 cells was measured in the presence of the P-glycoprotein efflux

inhibitor CP-100356 due to the high expression level of P-glycoprotein in this cell line[14]. Compound **2** dose-dependently protects the cell from death by blocking the replication of SARS-CoV-2 Delta, and the resulting half-maximal effective concentration (EC$_{50}$$^{CP}$) value was 319 nM (Fig. 1c, Supplementary Fig. 5). By comparison, the determined EC$_{50}$ of compound **2** was 7404 nM in the absence of CP-100356 (Supplementary Fig. 5). Moreover, the toxicity of **2** to Vero E6 cells was determined by CCK8 assay, and the resulting half-maximal cytotoxic

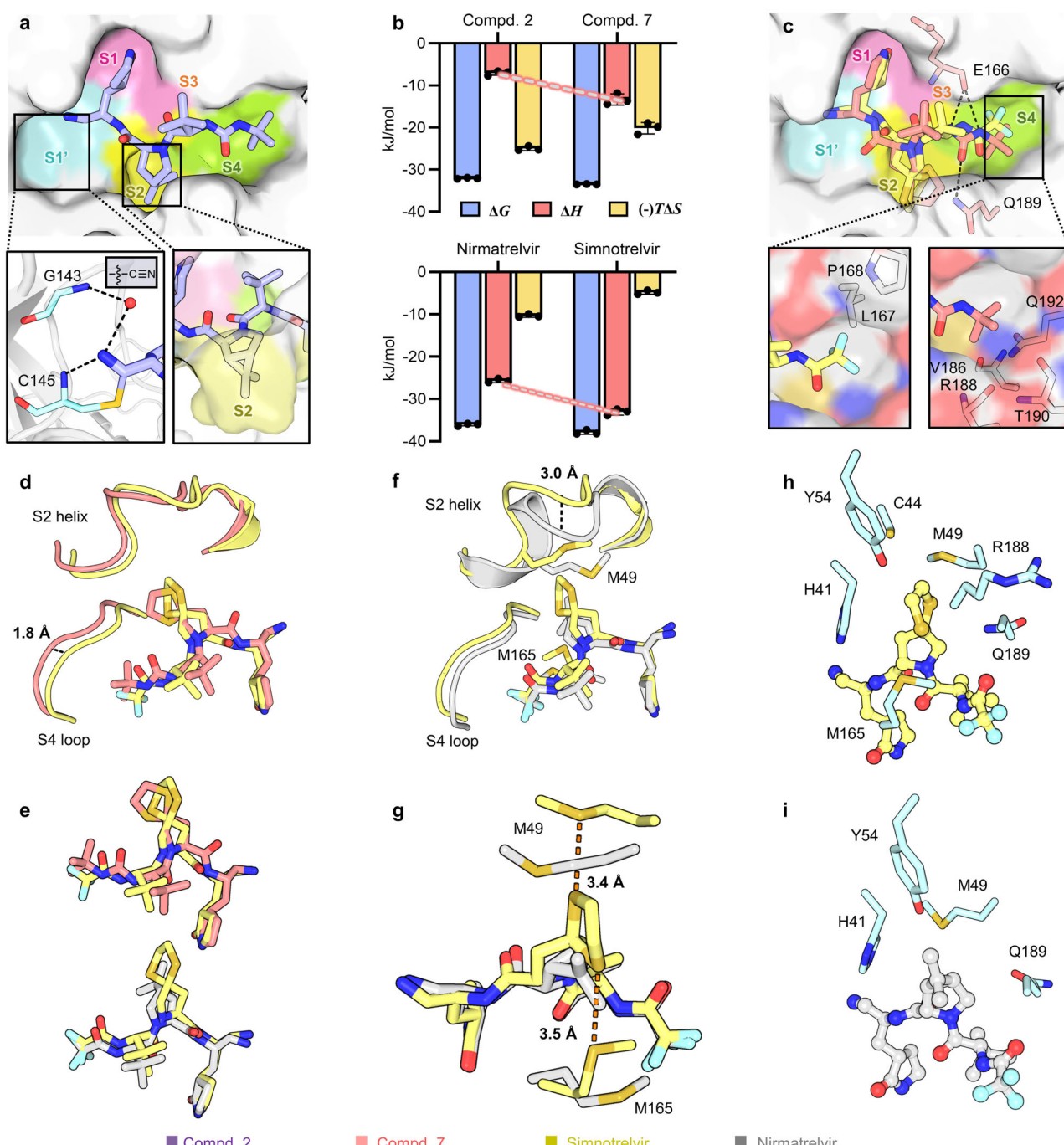

**Fig. 2 | Crystal structures and thermodynamic profiles of the inhibitors binding to SARS-CoV-2 3CL<sup>pro</sup>.** **a** Binding mode of compound **2** (light purple sticks, PDB code: 8IFQ) with SARS-CoV-2 3CL<sup>pro</sup>. The protease is shown in surface representation. The S1'-S4 subsites are colored as follows: S1', cyan; S1, magenta; S2, yellow; S3, orange; and S4, green. **b** Thermodynamic measurements of compounds **2** and **7**, nirmatrelvir, and simnotrelvir binding to SARS-CoV-2 3CL<sup>pro</sup>. The thermodynamic profiles are shown as histograms with the Gibbs free energy of binding ($\Delta G$; light blue), enthalpy ($\Delta H$; salmon) and entropy ($-T\Delta S$; light orange). The data are plotted as the mean ± SD from three independent experiments. **c–g** Comparison of the binding modes of compound **7** (salmon sticks, PDB code: 8IFS), simnotrelvir (yellow sticks, PDB code: 8IGX) and nirmatrelvir (gray sticks, PDB code: 8IGY) with SARS-CoV-2 3CL<sup>pro</sup> by superimposition of their crystal structures. Residues 40-51 (S2 helix) and 187-192 (S4 loop) of SARS-CoV-2 3CL<sup>pro</sup> are shown as cartoons. The distances are denoted as dashed lines. **h, i** Interactions between the P2 segment of simnotrelvir (**h**) and nirmatrelvir (**i**) (balls and sticks) and the S2 subsite residues (cyan sticks). Source data are provided as a Source Data file.

concentration (CC<sub>50</sub>) was over 500 μM (Fig. 1c, Supplementary Fig. 6), demonstrating the very low cytotoxicity of the compound. Based on these data, compound **2** appears to be a high-quality lead molecule for further development.

The crystal structures of SARS-CoV-2 3CL<sup>pro</sup> in complex with compounds **1–3** and boceprevir reveal that the P2-dimethyl cyclopropyl-proline in these compounds did not fully occupy the lipophilic S2 subsite (Fig. 2a). Substituents sizably larger than dimethyl cyclopropyl-proline, such as phenoxy-proline (**4**), phenyl-proline (**5**), phenylthiol-proline (**6**), dithiaspiro-proline (**7**), and bicyclo-proline (**8**), were thereby introduced at the P2 position of compound **2** during the second round of potency optimization, yielding five congeners, **4–8** (Fig. 1d). The resulting IC<sub>50</sub> values of these compounds were 6943 nM (**4**), 814 nM (**5**), 7215 nM (**6**), 23 nM (**7**), and 50 nM (**8**), with an ~310-fold

increase in the $IC_{50}$ value of the most potent compound (**7**) over the weakest compound (**6**) (Fig. 1d, Supplementary Fig. 1). It is thus implied that the P2-dithiaspiro-proline and the P2-bicyclo-proline groups in compounds **7** and **8**, respectively, might be a perfect match for the S2 subsite, although all five substituents seem to be accepted by this elastic subsite. In addition, **7** was also active in the cell-based antiviral assay with an $EC_{50}^{CP}$ value of 220 nM, making it more potent than compound **2**. Therefore, it is possible to put a group larger than dimethyl cyclopropyl-proline at the P2 position of the inhibitor to gain higher potency as well.

As the concentration of recombinant SARS-CoV-2 $3CL^{pro}$ used in the FRET-based enzymatic assay was 10 nM, the inhibitory ability of compounds with $IC_{50}$ values approaching 10 nM may not be faithfully reflected in this assay. The dissociation constant ($K_d$) of the inhibitors from the C145G mutant $3CL^{pro}$ measured by isothermal titration calorimetry (ITC) was therefore additionally used to evaluate the binding affinity of the inhibitor to the protease (expressed in $\Delta G = -RT\ln K_d$) (Fig. 1). Because the contribution of covalent bond formation between the warhead and C145 to the binding affinity is usually much higher than that of noncovalent interactions, C145 was mutated to glycine so that only the noncovalent binding affinity was explicitly determined. In contrast to the order of $IC_{50}$ values for the two compounds, the $K_d$ value of **7** versus **2** was 1683 nM versus 2968 nM, suggesting that replacement of dimethyl cyclopropyl-proline by dithiaspiro-proline at the P2 position improves the non-covalent binding affinity of the inhibitor to SARS-CoV-2 $3CL^{pro}$. More-over, the enthalpy–entropy profiles (expressed in $\Delta G = \Delta H - T\Delta S$) revealed by the ITC measurements also showed that the enthalpic gain upon the binding of compound **7** or **2** to SARS-CoV-2 $3CL^{pro}$ (C145G) was much lower than the entropic contribution (Fig. 2b), prompting us to conduct further optimization to achieve enthalpy-driven inhibitors with high affinity as well as selectivity[15–17].

Inspired by the discovery of PF-07321332 (nirmatrelvir) as an oral SARS-CoV-2 $3CL^{pro}$ inhibitor clinical candidate for the treatment of COVID-19[4], we focused our attention on P4 optimization. Four more compounds with various P4 segments, including P4-trifluoroacetyl (**9**), P4-1-fluorocyclopropane-1-acetyl (**10**), P4-methylsulfonyl (**11**), and P4-cyclopropanesulfonyl groups (**12**), derived from compound **7** were designed, synthesized and biochemically evaluated (Fig. 1e). The $IC_{50}$ values or dissociation constants of the four compounds were similar, ranging from 7 to 10 nM for $IC_{50}$ and 302 to 680 nM for $K_d$, demon-strating the stronger binding affinity and inhibition of SARS-CoV-2 $3CL^{pro}$ by these compounds over others. Importantly, compounds **9**–**12** also exhibited robust in vitro antiviral activity with nanomolar or low micromolar $EC_{50}^{CP}$ and $EC_{50}$ values in Vero E6 cells in the presence and absence of the P-glycoprotein inhibitor respectively (Fig. 1e, Sup-plementary Fig. 5). Among them, compound **9** (simnotrelvir) bearing the P4-trifluoroacetyl substituent was the most potent, with $EC_{50}^{CP}$ values of 26, 34, and 43 nM against the SARS-CoV-2 WIV04, Delta and Omicron strains, respectively (Supplementary Fig. 5). Negligible cyto-toxicity ($CC_{50} > 500$ μM) was observed for **9** in the same cell line (Supplementary Fig. 6). By comparison, an $EC_{50}^{CP}$ value of 30 nM was obtained for nirmatrelvir against the SARS-CoV-2 Delta strain in these cells (Supplementary Fig. 5). Additionally, the reversibility of SARS-CoV-2 $3CL^{pro}$ inhibition by compound **9** was evaluated, with irrever-sible inhibitors **3** and N3[18] as reference compounds. It demonstrated that a reversible and irreversible binding events occurred with **9** and **3** or N3, respectively (Supplementary Fig. 7). In other words, the nitrile in **9** forms a reversible covalent bond to the catalytic C145, while the α,β-unsaturated ketone in **3** or N3 irreversibly links to C145.

The noncovalent binding affinities of compounds **9**-**12** to SARS-CoV-2 $3CL^{pro}$ (C145G), as reflected by the $K_d$ values, were significantly improved compared to that of compound **7** (Fig. 1d, e). Remarkably, the enthalpy–entropy profile accompanying simnotrelvir (**9**) binding to the protease shows that it is a desired enthalpy-driven binding

inhibitor (Fig. 2b). After overlaying the crystal structures of SARS-CoV-2 $3CL^{pro}$ in complex with **7** and **9**, it was found that the introduction of the P4-trifluoroacetyl instead of the P4-tert-butylamide results in a shift of the binding positions of the P1-P3 segments of **9** as well as con-formational changes of the S4 loop and S2 helix in the SARS-CoV-2 $3CL^{pro}$/**9** complex compared to the SARS-CoV-2 $3CL^{pro}$/**7** complex (Fig. 2c–e). The overall shorter length along the peptidomimetic backbone because of a smaller P4 group allows **9** to easily fit into the substrate binding cavity. In contrast, compound **7** with the larger P4 group has to adopt a constrained conformation and barely squeezes into the cavity (Fig. 2c, d). This enables the hydrophobic P4-$CF_3$ group of **9** to favorably turn toward the hydrophobic region of the S4 subsite, whereas the P4-tert-butyl group of **7** has to orient toward the hydro-philic wall of the S4 subsite (Fig. 2c). This interaction also forces the S4 loop and the S2 helix to move toward **9** to form desirable contacts (Fig. 2d). Therefore, the significant change in the enthalpy–entropy profile of **9** compared to that of **7** is associated with the more relaxed binding conformation of **9** in which the positions of the P2 and P4 groups and three amide groups are optimal to form favorable inter-actions with the protease. Taken together, these data demonstrate that the introduction of a shorter and smaller P4-segment to replace the P4-tert-butylamide moiety not only results in improvement in the binding affinity or inhibitory activity but also changes the thermodynamic signature of protease-inhibitor binding.

Similar to simnotrelvir, the noncovalent binding of nirmatrelvir to SARS-CoV-2 $3CL^{pro}$ (C145G) is predominantly driven by enthalpic effects, with slightly lower binding affinity ($K_d = 620$ nM) than simno-trelvir ($K_d = 302$ nM) (Fig. 1e, Supplementary Table 1). The higher bind-ing affinity of simnotrelvir over nirmatrelvir is attributed to the increased enthalpic gain derived from the larger P2 segment in sim-notrelvir, which is evidenced by the crystal structures of SARS-CoV-2 $3CL^{pro}$ in complex with the two inhibitors (Fig. 2e–i, Supplementary Fig. 8). For direct comparison, the crystal structure of nirmatrelvir-bound SARS-CoV-2 $3CL^{pro}$ was also determined. The overall binding modes of simnotrelvir and nirmatrelvir are similar, but significant con-formational changes occur at the S2 subsite (Fig. 2f–g, Supplementary Figs. 8-9). The P2-dithiaspiro-proline group of simnotrelvir and its congener **10** enlarges the S2 subsite space by pushing away the M49-located loop, but additionally, two sulfur atoms establish S–S interac-tions at a distance of approximately 3.5 angstroms to M49 as well as M165, leading to the distinct movement of these two methionine resi-dues. As a result, more hydrophobic interactions are formed between the P2 segment of simnotrelvir and the surrounding residues, including M165, H41, Y54, C44, M49, R188, and Q189 (Fig. 2h), while in the case of nirmatrelvir, the P2-dimethyl cyclopropyl-proline moiety engages with only four residues, H41, Y54, M49, and Q189 (Fig. 2i). In order to gain a deep insight into the relationship between the enthalpic contribution ($\Delta H$) and hydrophobic interactions, the value of $\Delta Cp$ ($\Delta Cp = d\Delta H/dT$) of SARS-CoV-2 $3CL^{pro}$ binding with simnotrelvir and nirmatrelvir were further determined by ITC experiments at 20 °C, 30 °C, and 37 °C (body temperature), respectively. The resulted $\Delta Cp$ (−0.63 kJ mol$^{-1}$ K$^{-1}$) for the formation of the SARS-CoV-2 $3CL^{pro}$–simnotrelvir complex is more negative than that of the SARS-CoV-2 $3CL^{pro}$–nirmatrelvir complex (−0.41 kJ mol$^{-1}$ K$^{-1}$) (Supplementary Fig. 10). This indicates that the binding of simnotrelvir with SARS-CoV-2 $3CL^{pro}$ leads to the burial of more hydrophobic surface around the inhibitor compared to nirma-trelvir, in accordance with the results revealed by two crystal structures. Collectively, the biophysical, structural, and biochemical data demon-strate that the specific interaction pattern of the unique P2 and P4 segments with SARS-CoV-2 $3CL^{pro}$ results in the enthalpically more favorable binding signature of simnotrelvir compared to the other compounds, such as **7** and nirmatrelvir.

Ultimately, the X-ray crystal structure of SARS-CoV-2 $3CL^{pro}$ in complex with simnotrelvir allows for visualization of its mode of action in detail: (1) the nitrile warhead forms a covalent thioimidate ester

conjugate with the catalytic C145 and is also stabilized by H-bonds with the residues in the oxyanion hole; (2) the P1-5-membered lactam fits well into the S1 subsite, making favorable H-bonds with H163/E166/F140; (3) the induced-fit binding of the unique P2-dithiaspiro-proline to the S2 subsite results in extensive hydrophobic interactions with surrounding residues, particularly the specific S–S interactions with M49/M165; (4) the hydrophobic P3-*tert*-butyl group orients toward the bulk solvent and does not form any interactions with residues; (5) the small hydrophobic P4-trifluoroacetyl group is well suited to the S4 subsite to establish hydrophobic contacts; and (6) three backbone amide groups of simnotrelvir make three separate H-bonds to the main chains of E166/H164 (Supplementary Fig. 11). These integrated protein–ligand interactions enable simnotrelvir to covalently associate with the active site cavity of SARS-CoV-2 3CL$^{pro}$ and consequently potently inhibit the hydrolytic activity of the protease, providing a firm structural base to achieve excellent target engagement and robust antiviral efficacy in both preclinical and clinical evaluations for the treatment of COVID-19.

## Favorable PK and safety profiles of simnotrelvir

The outstanding in vitro antiviral activity together with the low cytotoxicity of simnotrelvir (**9**) prompted us to embark on more experiments to investigate its in vivo efficacy against viral infections. To this end, the in vivo pharmacokinetic properties of simnotrelvir were first assessed in rats (15 mg/kg IV, 15 mg/kg PO) and monkeys (5 mg/kg IV, 5 mg/kg PO). Simnotrelvir showed good oral bioavailability (*F*) in both rats (35.3%) and cynomolgus monkeys (41.9%), while it exhibited moderate to high plasma clearance with elimination half-lives ($t_{1/2}$) of 0.45 h and 1.7 h in rats and cynomolgus monkeys, respectively (Supplementary Table 2). Given the rapid metabolic turnover and that CYP3A was identified as the predominant CYP involved in metabolism (87% calculated metabolic contribution to simnotrelvir), we further investigated the effect of simnotrelvir in combination with ritonavir as a CYP3A inhibitor on drug plasma exposure in rats and monkeys (Fig. 3a, b, Supplementary Table 3). Simnotrelvir (5 mg/kg) in combination with 15 mg/kg ritonavir significantly increased the $C_{max}$ and $AUC_{0-\infty}$ by 3.9- and 9.6-fold in monkeys, respectively (Fig. 3b, Supplementary Table 3). A similar trend was also found in rats (Fig. 3a, Supplementary Table 3). These results suggest that simnotrelvir in combination with ritonavir achieves more adequate plasma exposure, which warrants further evaluation of the in vivo efficacy against SARS-CoV-2.

Simnotrelvir displays no significant inhibition against CYP enzymes with $IC_{50} \geq 30\,\mu M$ (Fig. 3c), suggesting a low risk of drug–drug interactions (DDIs). We also evaluated the selectivity of simnotrelvir against some human proteins. The results showed that simnotrelvir has no significant inhibitory activities against chymotrypsin at 20 μM, cathepsins A/D/E/L at 100 μM and 413 human kinases at 10 μM. The weak inhibition of simnotrelvir on cathepsin B ($IC_{50}$: 7194 nM) was observed, but the selectivity index (SI) was up to 799. Moreover, 100 μM simnotrelvir showed no inhibitory or stimulatory effects on the 47 targets that are well-known contributors to clinical adverse drug reactions (ADRs)[19]. Therefore, the high selectivity of simnotrelvir results in a good in vitro safety profile. In contrast, simnotrelvir displays high potency ($IC_{50}$: 8–13 nM) against six mutated 3CL$^{pro}$s (G15S, T21I, L89F, K90R, P132H, and L205V) that have been found in newly emerged SARS-CoV-2 variants, such as 3CL$^{pro}$ (P132H) in the Omicron variant (Fig. 3c, Supplementary Fig. 12). Simnotrelvir also effectively inhibited the activity of 3CL$^{pro}$ from six human CoVs including SARS-CoV ($IC_{50}$: 24 nM), MERS-CoV ($IC_{50}$: 60 nM), HKU1-CoV ($IC_{50}$: 10 nM), OC43-CoV ($IC_{50}$: 5 nM), H229E-CoV ($IC_{50}$: 129 nM), and NL63-CoV ($IC_{50}$: 849 nM) (Fig. 3c, Supplementary Fig. 12). These data show that simnotrelvir acts as a pan-CoV 3CL$^{pro}$ inhibitor with high off-target selectivity. In addition, no toxicity was observed for simnotrelvir at concentrations up to 150 μM to human peripheral blood mononuclear cells (PBMCs) and human hepatocytes. Simnotrelvir was also negative

in the in vitro Ames test, chromosome aberration test and in vivo micronucleus test in male rat bone marrow, indicating a low risk for genotoxicity; moreover, it has a low potential for cardiovascular risk, as indicated in the in vitro hERG inhibition test (Fig. 3c).

Simnotrelvir is also well tolerated and shows a good in vivo safety profile. Two-week GLP-compliant toxicity studies in rats (30-1000 mg/kg/day, Fig. 3d) and monkeys (40–600 mg/kg/day, Fig. 3e) were designed to assess the in vivo safety profile of simnotrelvir. As shown in Fig. 3d, e, the results demonstrated that simnotrelvir is well tolerated in both species and that exposure to simnotrelvir in plasma increases in a dose-dependent manner. The no observed adverse effect levels (NOAELs) were determined to be 1000 mg/kg/day for rats and 600 mg/kg/day for monkeys. The $C_{max}$ values in rats and monkeys at the NOAEL were 16,950 and 26,450 ng/mL, respectively. According to the plasma protein binding rate in rats (66.6%) and monkeys (80.7%), The $C_{max}$ values in rats and monkeys at the NOAEL are both >20-fold higher than the in vitro $EC_{90}^{total}$ (rat: 523 ng/mL, monkey: 906 ng/mL) against SARS-CoV-2 Delta replication, suggesting that no significant toxicity was observed at the target saturation dose.

## Robust in vivo antiviral activity protects lung and brain tissues from lesions

The in vivo antiviral activity of simnotrelvir was evaluated in a K18-hACE2 transgenic mouse model (Fig. 4a). After intranasal infection of male K18-hACE2 mice with the SARS-CoV-2 Delta strain (B.1.617.2), simnotrelvir at 50 mg/kg or 200 mg/kg plus 50 mg/kg ritonavir was orally administered to mice twice daily. Tissue samples were collected on Day 2 and Day 4 for measurement of the viral load and observation of histopathological changes. As shown in Fig. 4b, mice treated with simnotrelvir were protected from weight loss, while vehicle-treated mice showed ~10% weight loss, indicating that simnotrelvir protects mice from disease suffering and that no drug-related toxicity was observed under the continuous administration. On Day 2, compared with the vehicle group, simnotrelvir dose-dependently decreased the viral load in mouse lungs, represented by either viral titers (Fig. 4c) or by viral copy numbers (Supplementary Fig. 13a). A similar response was observed at Day 4 (Fig. 4d, Supplementary Fig. 13b). In addition, the viral antigen level in the lungs was detected by immunohistochemistry (IHC) analysis using an anti-SARS-CoV-2 nucleocapsid protein antibody (Supplementary Fig. 13c). Lungs from the vehicle mice showed strong staining, whereas the lungs from the mock mice were negative for staining. Simnotrelvir dose-dependently decreased the staining intensity, suggesting its ability to inhibit the virus replication in the lungs (Supplementary Fig. 13c). In addition to the lungs, SARS-CoV-2 Delta was found to attack the brain in K18-hACE2 mice. Viral replication in the brain seems to be delayed relative to that in the lungs, since the virus was hardly detected in vehicle mouse brains at Day 2 (Supplementary Fig. 13d) but quickly achieved a heavy load at Day 4 (Supplementary Fig. 13e). Similarly, simnotrelvir dose-dependently decreased the viral load in the brain, as represented by either viral titers (Fig. 4e) or viral copy numbers (Supplementary Fig. 13e). In comparison with uninfected mock mice, simnotrelvir treatment eliminated the virus from the brain (Fig. 4e, Supplementary Fig. 13e). Viral antigen staining assessed by IHC analysis also displayed an impressive decrease and again demonstrated the robust efficacy of simnotrelvir in the brain (Supplementary Fig. 13f).

We next explored the pathological changes histopathologically. The results revealed that simnotrelvir treatment protected the mouse lungs as well as brains from damage caused by SARS-CoV-2 Delta infection (Fig. 4f–i). The oral administration of simnotrelvir together with ritonavir significantly improved the alveolar atrophy or dilatation and alveolar septum thickening (Fig. 4f, h). In the brain, the pathological changes in the vehicle control group were mainly associated with neuron degeneration, and a few of the mice displayed bleeding. In the

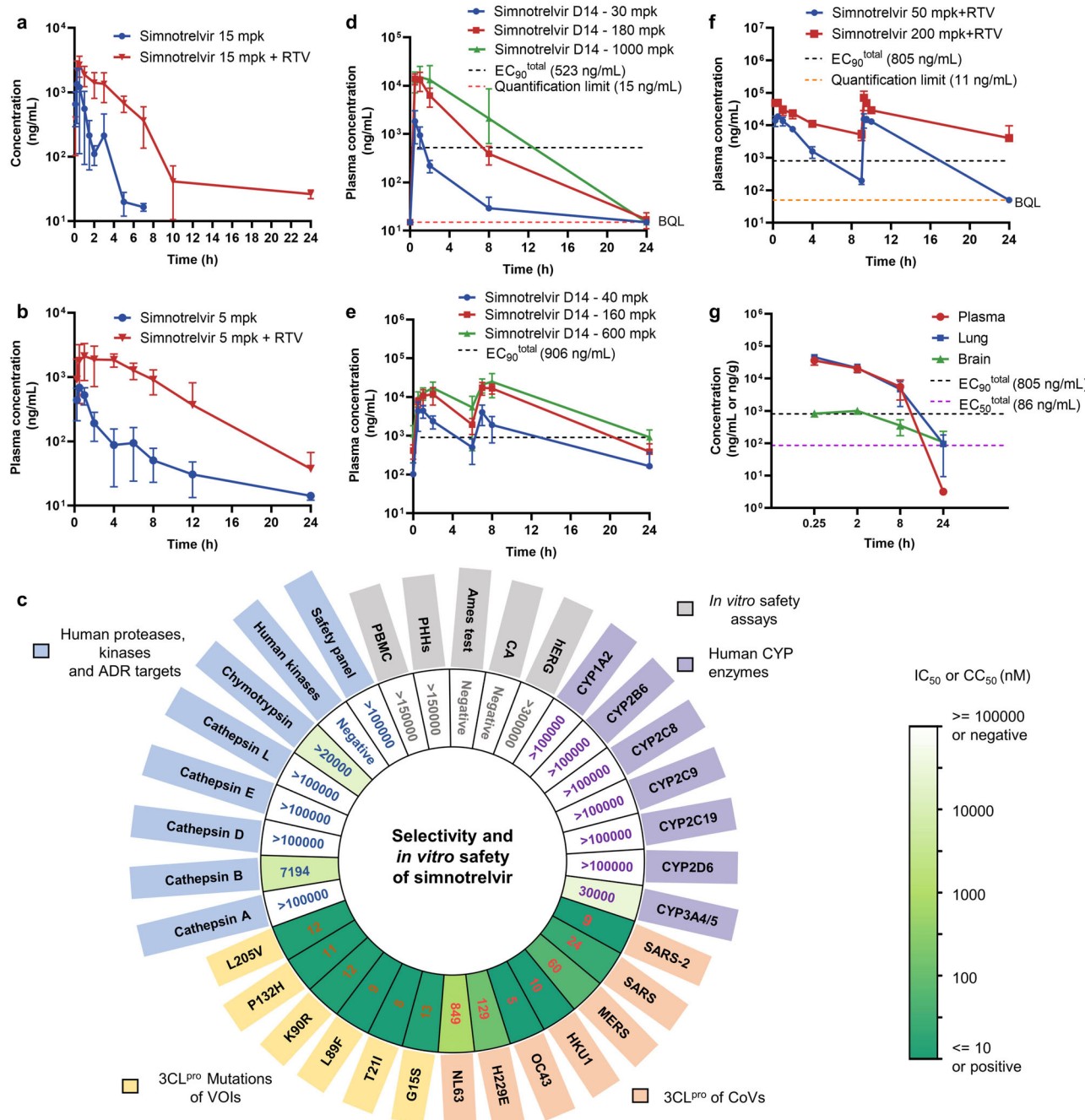

**Fig. 3 | Pharmacokinetics and safety profiles of simnotrelvir. a** Rat plasma concentration of simnotrelvir following a single oral dose of 15 mg/kg with or without 30 mg/kg ritonavir (RTV, $n = 6$ for each group). **b** Monkey plasma concentration of simnotrelvir following a single oral dose of 5 mg/kg with or without 15 mg/kg ritonavir ($n = 6$ for each group). **c** Heatmap of the measured $IC_{50}$ values of simnotrelvir against 6 human proteases, 413 human kinases and 47 human ADR targets (light blue); 5 safety assays (gray); 7 human CYP enzymes (light purple); 3CL$^{pro}$s of 7 human coronaviruses (light orange); and 3CL$^{pro}$s of 6 SARS-CoV-2 variants (light yellow). The different colors represent the measured $IC_{50}$ values or results (positive or negative). **d, e** Preclinical toxicokinetic exposure of rats (**d**) and monkeys (**e**) to simnotrelvir at Day 14. Rats were orally dosed with 30 mg/kg ($n = 8$), 180 mg/kg ($n = 8$) and 1000 mg/kg ($n = 8$) simnotrelvir once daily for 14 days.

Monkeys were orally dosed with 40 mg/kg (20 mg/kg, BID, $n = 6$), 160 mg/kg (80 mg/kg, BID, $n = 6$) and 600 mg/kg (300 mg/kg, BID, $n = 10$) simnotrelvir for 14 days. **f** Plasma exposure of uninfected male C57BL/J mice treated with simnotrelvir 200 mg/kg or 50 mg/kg plus 50 mg/kg ritonavir BID for one day ($n = 3$ for each time point). BQL indicates that the data were below the quantification limit. **g** Tissue distribution of simnotrelvir in uninfected male C57BL/J mice after a single oral administration of 200 mg/kg simnotrelvir plus 50 mg/kg ritonavir ($n = 24$ for each group). The unit of concentrations is ng/mL for plasma and ng/g for lung and brain tissues. The cellular $EC_{90}^{total}$ and $EC_{50}^{total}$ against SARS-CoV-2 Delta were calculated by dividing the cellular $EC_{90}^{CP}$ and $EC_{50}^{CP}$ by the plasma unbound fraction, respectively. The data are plotted as the mean ± SD. Source data are provided as a Source Data file.

group of mice treated with 200 mg/kg simnotrelvir, no bleeding was observed, and the prevention of neuron degeneration was confirmed, suggesting the protective effects of the compound on the brain (Fig. 4g, i).

By comparison, nirmatrelvir at 200 mg/kg plus 50 mg/kg ritonavir was orally administrated (BID) to K18-hACE2 mice after intranasal infection with SARS-CoV-2 Delta strain (B.1.617.2). Measurements of body weight loss and the viral load in lungs as well as brains

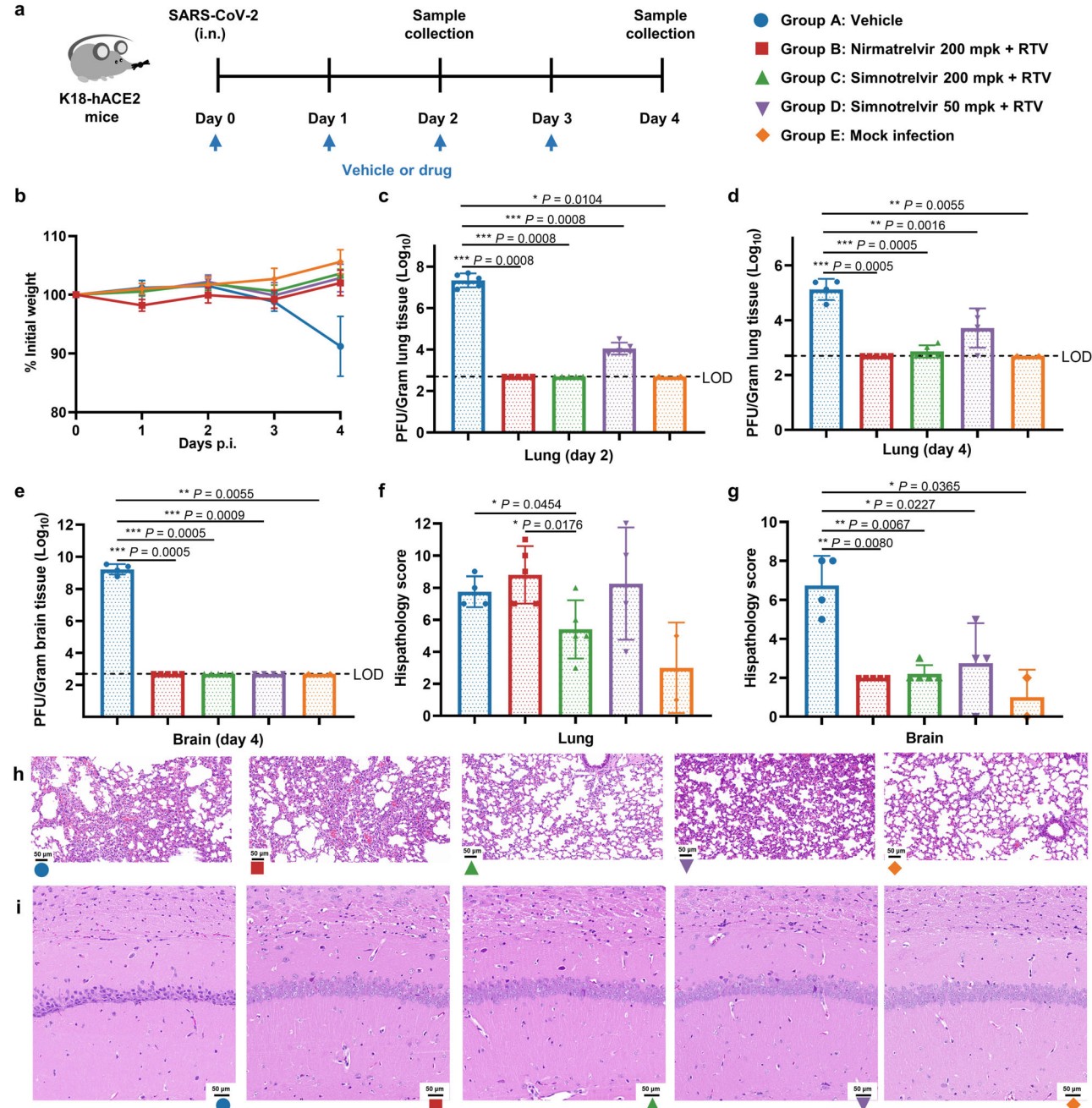

**Fig. 4 | In vivo oral efficacy of simnotrelvir in K18-hACE2 mice infected with the SARS-CoV-2 Delta strain. a** A flow chart of the K18-hACE2 mice infected with the SARS-CoV-2 Delta strain experiment. K18-hACE2 mice were divided into 5 groups: vehicle (infected control, blue circle, $n = 10$), nirmatrelvir 200 mg/kg plus ritonavir (RTV) 50 mg/kg (infected, red square, $n = 10$), simnotrelvir 200 mg/kg plus ritonavir 50 mg/kg (infected, green triangle, $n = 10$), simnotrelvir 50 mg/kg plus ritonavir 50 mg/kg (infected, purple inversed triangle, $n = 10$), and mock infection (uninfected control, orange diamond, $n = 2$). **b** Changes in mouse body weight were monitored daily ($n = 4, 5, 5, 4$, and 2 for group A–E). **c** Viral titers in lungs at Day 2 post-infection ($n = 5, 5, 5, 5$, and 2 for group A–E). **d** Viral titers in lungs at Day 4 post-infection ($n = 4, 5, 5, 4$, and 2 for group A-E). **e** Viral titers in brains at Day 4 post-infection ($n = 4, 5, 5, 4$, and 2 for group A–E). **f, g** Histopathology scores of lung (**f**) and brain (**g**) necrosis and confluent areas of inflammation. **h, i** H&E-stained sections of lungs (**h**) and brains (**i**) at Day 4 post-infection ($n = 4, 5, 5, 4$, and 2 for group A–E). The scale bars represent 50 μm. LOD indicates the lower limit of detection. The results of body weights and viral titers are graphed as the mean ± SD. Statistical differences were determined by one-way ANOVA in **c-e** and two-tailed Welch's $t$ test in (**f, g**). $^*P < 0.05$, $^{**}P < 0.01$, $^{***}P < 0.001$, $^{****}P < 0.0001$. Source data are provided as a Source Data file.

showed that the oral treatment with nirmatrelvir and simnotrelvir at the same dosage yielded comparable results (Fig. 4, Supplementary Fig. 13), suggesting that these two compounds have roughly identical antiviral activity in vivo. Nevertheless, it has been noted from the histopathological evaluation of the lungs that simnotrelvir had a better protective effect against viral infections than nirmatrelvir (Fig. 4f, h), although the mechanism by which simnotrelvir

significantly alleviates the pathological features in the lungs is currently unclear.

To explore the correlation between antiviral efficacy and drug exposure, we performed two independent PK studies to evaluate drug exposure in the plasma and brains under the efficacious dose. In naïve C57BL/J mice from which the K18-hACE2 mice were developed, simnotrelvir at a dosage of 50 mg/kg or 200 mg/kg plus 50 mg/kg

ritonavir was orally co-administered twice daily, resulting in a peak plasma concentration at 0.5 h and a $C_{max}$ value of 18500 or 49300 ng/mL, respectively. According to the plasma protein binding rate in mice (78.3%), the calculated drug concentrations are approximately 23- and 61-fold of the in vitro $EC_{90}^{total}$ (805 ng/mL) against the Delta variant. While the drug concentration was maintained above the $EC_{90}^{total}$ for 24 h at a dosage of 200 mg/kg (Fig. 3f). In another satellite group, a single dose of 200 mg/kg simnotrelvir plus 50 mg/kg ritonavir in naïve C57BL/J mice resulted in comparably high exposure in the lungs and plasma (Fig. 3g). Notably, moderate simnotrelvir exposure was observed in the mouse brains after the single dose of 200 mg/kg simnotrelvir plus 50 mg/kg ritonavir. Although it is difficult to completely avoid the contamination from the blood, the measured concentration of simnotrelvir in the brain is higher than that in the plasma at 24 h, demonstrating that simnotrelvir is distributed to the brain. It was shown that the peak drug concentration in the brains reached the in vitro $EC_{90}^{total}$ and covers $EC_{90}^{total}$ and $EC_{50}^{total}$ (86 ng/mL) for more than 2 h and 8 h, respectively. It was also calculated that oral administration of 200 mg/kg simnotrelvir plus 50 mg/kg ritonavir twice daily could maintain more than 90% inhibition of viral replication for at least 4 h. Taken together, these data demonstrated a good correlation between the drug exposure and antiviral efficacy in vivo for the combined oral administration of simnotrelvir and ritonavir.

## Discussion

Numerous drugs targeting viral proteases have successfully reached the market, such as saquinavir, boceprevir and nirmatrelvir. Further development of new viral protease inhibitors based on these marketed drugs has afforded a cutting-edge solution to efficiently find therapeutic interventions against emerging highly pathogenic viruses. Herein, we present our efforts in structure-based drug design and development of oral SARS-CoV-2 3CL^pro inhibitors: step-by-step medicinal chemistry optimization of the warhead and P1, P2, and P4 segments of boceprevir eventually led to the creation of the anti-COVID-19 drug simnotrelvir. Boceprevir has been widely used as a starting point for the design of viral protease inhibitors, including SARS-CoV-2 3CL^pro inhibitors[20,21], owing to its clinical application as an oral anti-HCV drug with a peptidomimetic chemical structure and a covalent mode of action. When we embarked on this drug discovery project to find an oral 3CL^pro inhibitor for the treatment of infection by SARS-CoV-2 and its emerging variants, boceprevir served as an attractive hit compound with favorable drug-like properties. The weak inhibition of boceprevir against SARS-CoV-2 3CL^pro and its imperfect binding mode to 3CL^pro shown in the cocrystal structure provided an opportunity for us to use structure-guided optimization to improve the potency. The inclusion of nitrile, the aldehyde and the α,β-unsaturated ketone groups that have been used in approved covalent drugs[22,23] was first investigated, resulting in identification of the nitrile as a promising moiety due to it having the lowest reactivity. Subsequently, optimal elements for the P1, P2, and P4 positions of the inhibitor to perfectly engage subsites S1, S2, and S4 of the protease were explored, leading to the discovery of the P1-5-membered lactam, P2-dithiaspiro-proline and P4-trifluoroacetyl groups for simnotrelvir. The P3-*tert*-leucine was inherited directly from boceprevir, as its side chain is fully exposed to the bulk solvent and may provide favorable entropic contributions to the binding affinity.

The simnotrelvir discovery process again shows the elegance and utility of structure-based drug design and its significance in facilitating the development of protease-specific inhibitors. The intrinsic flexibility of peptidomimetic compounds coupled with a high degree of malleability of the protease substrate binding cavity enable a broad spectrum of structurally diverse ligands with differing binding affinities to bind to the protease. This underscores the fact that the binding modes of the ligands in complex with the protease explicitly revealed in the X-ray crystal structures are fundamental to interpret the structure–activity relationships and to accelerate the rational design of more potent inhibitors. In addition, the protein–ligand binding process is controlled by thermodynamics. ITC measurements for thermodynamic profiling provide insights into the complicated relationships between enthalpy–entropy profiles and quantified protein–ligand interactions. A comprehensive understanding of the structure-activity relationships of a series of ligands toward the primary target protein based on an integration of the ligand binding enthalpy/entropy data and the protein–ligand interaction pattern precisely determined by high-quality crystal structures is thus critical to efficiently guide rational drug design and eventually lead to the identification of simnotrelvir with optimal potency.

Although developed independently from and in parallel with nirmatrelvir, which was primarily created on the basis of PF-00835231, the structure as well as the mode of action of simnotrelvir (derived from boceprevir utilizing a structure-based drug design strategy) is similar to that of nirmatrelvir. The in vitro and in vivo antiviral activities of the two drugs were also comparable. The unique feature distinguishing simnotrelvir from nirmatrelvir revealed by their crystal structures with the protease is that the P2-dithiaspiro-proline, a large and rare P2 segment, is employed for simnotrelvir, and more hydrophobic interactions are accordingly formed between the P2-dithiaspiro-proline and the induced-fit S2 subsite, which is accompanied by significant conformational changes of the S2-helix and M49/M165 (Fig. 2, Supplementary Figs. 8-9). This renders the noncovalent binding portion of simnotrelvir able to recognize SARS-CoV-2 3CL^pro with higher binding affinity than nirmatrelvir (Supplementary Table 1). Accordingly, the unconventional S–S interaction between dithiaspiro-proline and the side chains of two methionine residues (M49/M165) is unique for simnotrelvir among known 3CL^pro inhibitors. This major difference also provides a structural basis for improving the selectivity of simnotrelvir toward off-target proteins.

While introduction of a cyclic segment at P2 position is a key step to increase the oral bioavailability of peptidomimetic inhibitors, only three types (I-III) of cyclic segments have been found in crystal structures of SARS-CoV-2 3CL^pro in complex with 161 different peptidomimetic inhibitors. The major one is the dimethyl cyclopropyl-proline group that is also utilized by nirmatrelvir (Supplementary Fig. 14a). The dithiaspiro-proline group that has rarely been exploited for drug design was incorporated into simnotrelvir as a cyclic P2-segment fitting well into the S2 subsite of SARS-CoV-2 3CL^pro. To the best of our knowledge, this group has never been used as a segment for the protease inhibitor design before. Moreover, the dithiaspiro-proline group forms more hydrophobic interactions with the S2 subsite compared to other cyclic segments (Supplementary Fig. 14b, c), thereby enhancing the binding affinity of simnotrelvir to the protease. Accordingly, the approval of simnotrelvir provides a proof-of-concept guide to utilize the dithiaspiro-proline as a vital cyclic segment for the design of oral peptidomimetic inhibitors targeting 3CL^pros as well as other proteases.

In addition to the peptidomimetic inhibitors like nirmatrelvir and simnotrelvir, ensitrelvir (S-217622) is a non-peptidomimetic, non-covalent SARS-CoV-2 3CL^pro inhibitor that has been approved as an oral drug for the treatment of COVID-19 in Japan. It was identified mainly through the virtual screening followed by structure-guided optimization. The crystal structure reveals that ensitrelvir binds to the active site of SARS-CoV-2 3CL^pro too, but it only occupies S1', S1, and S2 subsites without the covalent link to the catalytic residue C145, distinct from the binding modes of the peptidomimetic inhibitors (Supplementary Fig. 9). Ensitrelvir is orally administered without ritonavir, that is beneficial for therapeutic treatment. According to these approved inhibitors, covalent inhibitors of 3CL^pro with less peptidomimetic feature are worth developing in future drug discovery.

In summary, we have created a series of electrophilic warhead-containing peptidomimetic inhibitors of SARS-CoV-2 3CL^pro based on boceprevir. The in vitro and in vivo antiviral activity studies together

with preclinical investigations ultimately identified compound **9** (simnotrelvir) as an oral clinical candidate. Simnotrelvir possesses high on-target affinity and off-target selectivity, broad-spectrum inhibitory activity against SARS-CoV-2 as well as other CoV 3CL^pros, and ability to effectively block SARS-CoV-2 and its variant replication in cells and mice; moreover, the binding mode and thermodynamic signature of simnotrelvir have been precisely determined. Simnotrelvir is also safe and exhibits sufficiently high oral plasma concentrations when administered alone or in combination with ritonavir, which is well correlated with the robust in vivo antiviral efficacy. These advantages facilitate simnotrelvir to successfully pass through both preclinical and clinical evaluations. The approved use of simnotrelvir in combination with ritonavir as an oral therapy not only offers an effective long-term treatment strategy for COVID-19 and other coronavirus diseases but also expands the clinical application and significance of covalent peptidomimetic inhibitors targeting viral proteases.

## Methods

### Chemistry

Procedure and synthetic schemes for compounds **1–12** are shown in Supplementary Figs. 15–18. Spectral data for compounds **1–12** are included in the Spectral Data for Synthetic Compounds section of the Supplementary Information. All commercially available chemicals and solvents were used directly without further purification. All reactions were monitored by thin layer chromatography (TLC) on silica gel plates (GF-254). All compounds were characterized by their NMR and MS spectra. $^1$H NMR and $^{13}$C NMR spectra were recorded on a Bruker Avance 800, a Bruker Avance 600, a Bruker Avance 500 or a Bruker Avance 400 using TMS as the internal standard, and $^{19}$F NMR spectra were recorded on a Bruker Avance 400 (376 MHz for $^{19}$F NMR). NMR data were recorded using TopSpin software (v3.0 or v3.2 or v3.5) and analyzed using MestReNova (v12.0.0-20080). HPLC spectra were collected and analyzed on an Agilent1100 system using Chemstation (vB.04.02) or an Agilent1260 Infinity II HPLC system using Chromeleon (v7.2 SR5). High-resolution mass spectrometry (HRMS) spectra were acquired with an Agilent G6520 Q-TOF mass spectrometer using MassHuter Workstation software (vB.05.01) and analyzed with Xcalibur software (v2.0.5). Low-resolution mass spectrometry (LRMS) spectra were acquired with a Thermo Fisher FINNIGAN LTQ spectrometer and analyzed using Xcalibur software (v2.0.5). The purities of compounds used in the biological tests were over 95%. Analytical analysis for purity was determined by the following methods. Method A: Agilent 1100, LC-SH-587 Ascentis RP-Amide 250 × 4.6 mm, 5 μm, using mobile phases A (water with 0.1% $H_3PO_4$ and B (CH$_3$CN). Gradient elution started with 80% A at 0 min, and the proportion of A decreased to 10% in 20 min. The flow rate was 1.0 mL/min, detection occurred at a wavelength of 210 nm, and the column oven temperature was 30 °C. Method B: Agilent 1260, LC-SH-587 UPELCO Ascentis RP-Amide 250 × 4.6 mm, 5 μm, using mobile phases A (water with 0.1% $H_3PO_4$ and B (CH$_3$CN). Gradient elution started with 60% A at 0 min, and the proportion of A decreased to 30% in 20 min. The flow rate was 1.0 mL/min, detection occurred at a wavelength of 210 nm, and the column oven temperature was 30 °C. Method C: Agilent 1100, LC-SH-425 Agilent ZORBAX SB-C18 150×4.6 mm, 5 μm, using mobile phases A (water with 0.1% $H_3PO_4$ and B (CH$_3$CN). Gradient elution started with 80% A at 0 min, and the proportion of A decreased to 10% in 20 min. The flow rate was 1.0 mL/min, detection occurred at a wavelength of 210 nm, and the column oven temperature was 30 °C. Method D: Agilent 1100, LC-SH-588 SUPELCO Ascentis RP-Amide 250 × 4.6 mm, 5 μm, using mobile phases A (water with 0.1% $H_3PO_4$ and B (CH$_3$CN). Gradient elution started with 70% A at 0 min, and the proportion of A decreased to 10% in 20 min. The flow rate was 1.0 mL/min, detection occurred at a wavelength of 210 nm, and the column oven temperature was 30 °C.

### Protein expression and purification

The cDNA of 3CL^pros of SARS-CoV-2 (GenBank: MN908947.3), SARS-CoV (GenBank: AAP13442.1), MERS-CoV (GenBank: MT387202.1), H229E-CoV (GenBank: AF304460.1), HKU1-CoV (GenBank: AY597011.2), NL63-CoV (GenBank: AY567487.2), and OC43-CoV (GenBank: AY903459.1) with an N-terminal SUMO tag were cloned into the pET-15b vector. The SARS-CoV-2 3CL^pro mutants, including G15S, T21I, L89F, K90R, P132H, C145G, and L205V, used for inhibition assays or ITC measurements were obtained according to the restriction-free method[24]. The plasmids were then transformed into BL21 (DE3) cells for protein expression. The expressed proteins were purified by Ni-NTA column chromatography (GE Healthcare) and cleaved by SUMO-specific peptidase 2 (SENP2) to remove the SUMO tag. The resulting protein samples were further purified by Q-Sepharose followed by size-exclusion chromatography (GE Healthcare). The eluted protein samples were stored in a solution of 10 mM Tris, pH 7.5 for the enzymatic inhibition assays and protein crystallization or in a solution of 50 mM Tris, pH 7.5 for ITC measurements.

### 3CL^pro enzymatic inhibition assays

The inhibitory activities of the compounds against 3CL^pros were determined by fluorescence resonance energy transfer (FRET) protease assays. The fluorogenic substrate DABCYL-KTSAVLQSGFRKME-EDANS (GenScript, China) can be cleaved by 3CL^pro and generates an EDANS peptide fragment that emits an intense fluorescence signal at excitation/emission wavelengths of 340/490 nm. The FRET-based enzymatic assay was performed as follows. Recombinant 3CL^pro was mixed with serial dilutions of each compound in 80 μL of assay buffer (50 mM Tris, pH 7.3, 1 mM EDTA) and incubated for 10 min at room temperature. The final concentrations of SARS-CoV-2, SARS-CoV, MERS-CoV, HKU1-CoV, OC43-CoV, H229E-CoV, and NL63-CoV 3CL^pros used in the assays were 10, 30, 80, 20, 10, 30, and 30 nM, respectively. The reaction was initiated by adding 40 μL of fluorogenic substrate to a final concentration of 10 μM. After that, the fluorescence signal at 340 nm (excitation)/490 nm (emission) was immediately measured every 1 min for 10-60 min with a microplate reader. The initial velocities of the reactions with added compounds compared to the reaction with DMSO were calculated and used to generate IC$_{50}$ curves. Three independent experiments, each carried out in triplicate, were performed to determine the IC$_{50}$ values of the compounds against SARS-CoV-2 3CL^pro. The IC$_{50}$ values are expressed as the mean ± SD. Experiments were performed in triplicate to determine the IC$_{50}$ values of simnotrelvir against SARS-CoV, MERS-CoV, HKU1-CoV, OC43-CoV, H229E-CoV, NL63-CoV 3CL^pros, and the SARS-CoV-2 3CL^pro mutants G15S, T21I, L89F, K90R, P132H, and L205V. At least nine concentrations of each compound were used to determine the IC$_{50}$ values. The IC$_{50}$ values were calculated by GraphPad Prism (v9.1.2).

### Enzymatic inhibition assays of mammalian proteases

Bovine chymotrypsin (5 nM), human cathepsin A (25 nM), cathepsin B (2 nM), cathepsin D (10 nM), cathepsin E (0.2 nM), and cathepsin L (20 nM) were added to plates containing compounds and incubated at room temperature for 10-30 min. The reactions were started by the addition of 10 μM chymotrypsin substrate Suc-AAPF-AMC, 50 μM cathepsin A substrate MCA-RPPGFSAFK(DNP)-OH, 2 μM cathepsin B substrate Z-RR-AMC, 30 μM cathepsin D substrate MCA-PLGL-Dpa-AR-NH2, 10 μM cathepsin E substrate MOCAc-GKPILFFRLK(DNP)DR-NH2, or 20 μM cathepsin L substrate Z-FR-AMC. The reactions proceeded for approximately 10-60 min and were monitored by a microplate reader. The reaction rates of the well containing compounds were compared to that of the well containing DMSO to determine the inhibition ratio. Experiments were performed in triplicate. The IC$_{50}$ values were calculated by GraphPad Prism (v9.1.2).

## Reversibility evaluation of SARS-CoV-2 3CL^pro inhibition by the compounds

To evaluate the reversibility of SARS-CoV-2 3CL^pro inhibition by the compounds, SARS-CoV-2 3CL^pro at a final concentration of 2 μM was incubated with 2 μM simnotrelvir, **3**, or N3 at 4 °C for 1 h. N3 is a known covalent inhibitor of 3CL^pro with an α,β-unsaturated ketone warhead and was thus used as a control. The reaction was initiated by adding 40 μL of fluorogenic substrate at a final concentration of 10 μM. The fluorescence signal at 340 nm (excitation)/490 nm (emission) of each of the incubation mixtures as well as that after 50- and 200-fold dilution were immediately measured every 20 s for 3 min. The initial velocities of reactions with the added compound after 50- and 200-fold dilution compared to the reaction with DMSO were calculated to generate the ratio of fractional velocity. Three independent experiments (each experiment in triplicate) were carried out. The values are expressed as the mean ± SD.

## Half-life determination of the compounds reacting with GSH

To determine the rates of the compounds reacting with GSH (GSH$t_{1/2}$), compounds **1**, **2**, and **3** were incubated with 10 μM GSH at 37 °C in potassium phosphate buffer, pH 7.4 (1% DMSO). 1H-Pyrrole-2,5-dione (CPM, ThermoFisher) at a concentration of 50 μM was added to the reaction system at predetermined times to quantify the remaining GSH. After that, the fluorescence signal at 384 nm (excitation)/470 nm (emission) was immediately measured using a microplate reader. The GSH$t_{1/2}$ was derived from the semilogarithmic plots of the relative amount of GSH remaining versus incubation time using GraphPad Prism (v9.1.2). Three independent experiments (each experiment in duplicate) were performed. The values are expressed as the mean ± SD.

## ITC measurements

All measurements were performed in ITC buffer containing 50 mM Tris (pH 7.5) with an iTC$_{200}$ calorimeter (General Electric Co.). The compounds were diluted in ITC buffer to final concentrations of 0.5-1 mM. The purified SARS-CoV-2 3CL^pro C145G was diluted in ITC buffer to final concentrations of 0.05-0.1 mM. The final concentration of DMSO in the reaction was less than 1%. All titrations were performed using an initial injection of 0.4 μL followed by 19 identical injections of 2 μL each with a duration of 4 s per injection and a spacing of 120 s between injections. The last five data points were averaged as the heat of dilution and subtracted from each titration. ITC experiments were conducted at 30 °C to obtain the thermodynamic binding signature of each compound with SARS-CoV-2 3CL^pro C145G. To measure the Δ$Cp$ values of simnotrelvir and nirmatrelvir binding to SARS-CoV-2 3CL^pro C145G, ITC experiments were also performed at 20 °C and 37 °C. Three independent experiments were carried out. The values are expressed as the mean ± SD of three independent experiments.

## Protein crystallization and structure determination

The purified SARS-CoV-2 3CL^pro protein was concentrated to 7 mg/mL for crystallization. To obtain complex structures, the SARS-CoV-2 3CL^pro protein was incubated with 2–4 mM inhibitors for 1 h before crystallization. Crystals of the complex were obtained under the conditions of 10-22% PEG6000, 100 mM MES, pH 5.75-6.25, and 3% DMSO. Crystals were flash-frozen in liquid nitrogen in the presence of the reservoir solution supplemented with 20% glycerol. X-ray diffraction data were collected at beamlines BL02U1 (BL17U1), BL18U1, and BL19U1 at the Shanghai Synchrotron Radiation Facility[25,26]. The data were processed with HKL3000 software packages[27]. The complex structures were solved by molecular replacement using the program CCP4 (v7.0.078)[28] with a search model of PDB code 6M2N[29]. The model was built using COOT (v0.8.9.2)[30] and refined with the program PHENIX (v1.17.1-3660)[31]. The complex structures were analyzed with Pymol (v2.4.0) and LigPlot (v2.2.8). The refined structures were deposited in the Protein Data Bank with the accession codes

listed in Supplementary Table. 4. The complete statistical data as well as the quality of the solved structures are also shown in Supplementary Table 4.

## CYP inhibition assay

Human liver microsomes (Corning Corp, 6123001) were used to evaluate the potential of simnotrelvir to inhibit cytochrome P450s. The microsomes were incubated with specific probe substrates and simnotrelvir at various concentrations (0 ~ 100 μM). All incubations were performed at 37 °C in a water bath. The enzymes were carefully thawed on ice prior to the experiment. After preincubation for 5 min, the reactions were initiated by the addition of 1 mM NADPH. Reactions were terminated at predetermined time points by adding an equal volume of ice-cold acetonitrile. Subsequently, the activities of CYP enzymes were determined by LC-MS/MS methods. The data were analyzed using Analyst (v1.7.1) and GraphPad Prism (v9.1.2). Experiments were performed in duplicate.

## Selectivity evaluation of simnotrelvir against kinases

The selectivity of simnotrelvir against 413 human kinases was determined by using Eurofins standard KinaseProfiler assays (Eurofins Cerep, 50-005KP10). Protein kinases, with the exception of ATM (h) and DNA-PK (h), were assayed in a radiometric format, whereas lipid kinases, ATM (h), ATR/ATRIP (h), and DNA-PK (h), were assayed with a time-resolved fluorescence (HTRF) format. Simnotrelvir at a final concentration of 10 μM was mixed with the kinase and substrate at 37 °C. The reaction was initiated by the addition of ATP at a final concentration of 10 μM. Then, the signal of each well was measured with a microplate reader and compared to that of the DMSO reference. The remaining kinase activity was calculated according to the formula (1) and (2) shown below. Experiments were performed in duplicate.

For radiometric assays:

$$\text{The remaining activity} = \frac{\text{mean of sample counts} - \text{mean of blank counts}}{\text{mean of control counts} - \text{mean of blank counts}} \quad (1)$$

For HTRF assays:

$$\text{The remaining activity} = \frac{\text{mean of sample counts} - \text{mean of no enzyme counts}}{\text{mean of control counts} - \text{mean of no enzyme counts}}$$

$$(2)$$

## Selectivity evaluation of simnotrelvir against safety related targets

The SAFETYscan assay (Eurofins Cerep, 87-1003DR) was used to assess the selectivity against safety-related targets, including GPCRs, nuclear hormone receptors, ion channels, transporters, kinases and enzymes. The experiment was carried out by following Eurofins's relevant standard operating procedures at 37 °C. To monitor the activation or inhibition of simnotrelvir on GPCRs, the hit hunter cAMP assay and calcium No washPLUS assay were applied. PathHunter NHR protein interaction (NHR Pro) and nuclear translocation (NHR NT) assays were used to assess the activation of a nuclear hormone receptor with enzyme fragment complementation. To evaluate the effect of simnotrelvir on ion channels and transporter proteins, three assays, including the FLIPR potassium assay, the calcium NWPLUS assay and the FLIPR membrane potential assay, were used. The KINOMEscan screening platform was employed to quantitatively measure the interactions between the test compound and kinases. Enzymatic assays were used to determine simnotrelvir inhibition of enzymatic activity by measuring either the consumption of substrate or production of product over time. Ten concentrations (up to 100 μM) of simnotrelvir were used to evaluate the selectivity of simnotrelvir against each safety-related target. Experiments were performed in duplicate.

## In vitro Ames test

The objective of this assay was to evaluate the mutagenicity of simnotrelvir following OECD guideline TG471 and ICH guideline S2 (R1). Histidine-dependent Salmonella typhimurium (TA97a, TA98, TA100, and TA1535) and tryptophan-dependent Escherichia coli (WP2) were obtained from Molecular Toxicology Inc. and used as test strains. The test was conducted using the plate incorporation method in the absence (-S9) and presence (+S9) of the metabolic activation system (rat liver S9 obtained from Molecular Toxicology Inc., batch No. 4505). DMSO was used in the test as the vehicle. The doses of simnotrelvir selected for this study were 15, 50, 150, 500, 1500, and 5000 µg/plate in the absence and presence of S9. Concurrent negative and positive control groups were included. During the test, test article precipitation, background lawn growth and the number of revertant colonies were evaluated. The mutagenicity of the test article was determined as1 (negative) if the mean number of revertant colonies in all dose groups was less than twice that in the negative control group; 2 (positive) if the mean number of revertant colonies in any dose group was more than 4 times that in the negative control group; and 3 (equivocal positive) if the mean number of revertant colonies in the test article group exhibiting a maximum response was 2 to 4 times that in the negative control group, in which case a final conclusion was drawn through a comprehensive analysis using SPSS Statistics (v21) and biological significance. Experiments were performed in triplicate.

## Chromosome aberration test

The objective of this study was to evaluate the potential of simnotrelvir to induce chromosome damage by detecting the frequency of structural chromosomal aberrations in Chinese hamster lung (CHL) cells (ATCC, CRL-1935) following OECD guideline TG473 and ICH guideline S2 (R1). This study was conducted both in the absence (-S9) and in the presence of a metabolic activation system (+S9) (rat liver S9 obtained from Molecular Toxicology Inc., batch No. 4471). In the absence of the metabolic activation system (−S9), CHL cells were treated with simnotrelvir for approximately 4 h and 24 h (−S9 4 h and −S9 24 h). In the presence of the metabolic activation system ( +S9), CHL cells were treated with simnotrelvir for approximately 4 h (+S9 4 h). Three doses of simnotrelvir at 125, 250, and 500 µg/mL were used for each test condition in this study. Concurrently, negative and positive control groups were included for each test condition. Precipitation, cytotoxicity, and chromosome (including chromatid type) structural aberrations (excluding gaps) were observed and evaluated under each condition. The data were analyzed using SPSS Statistics (v21).

## In vivo micronucleus test in male rat bone marrow

The objective of this study was to identify the potential of simnotrelvir to induce micronucleated polychromatic erythrocytes (PCEs) in the bone marrow of Sprague Dawley (SD) rats (SPF grade, Laboratory animal sales department, Shanghai Institute Of Planned Parenthood Research) after 2 consecutive days by oral gavage. All procedures related to animal handling, care and treatment in the rat bone marrow micronucleus test were performed according to approved guidelines. Thirty male SD rats (6 animals/group, 5 groups in total) were given simnotrelvir (500, 1000, and 2000 mg/kg), cyclophosphamide monohydrate (CP, 20 mg/kg, batch No. WXBD5120V, Sigma–Aldrich Co., LLC.) or the control formulation via oral gavage. In the control formulation and test article groups, the animals were treated once daily for 2 consecutive days, and bone marrow smears were prepared from the animals euthanized 18 h to 24 h after the last dosing. In the CP group, the animals were dosed once on Day 2, and bone marrow smears were prepared from the animals euthanized 24 h to 30 h after dosing. During the study, clinical observations, body weight, the ratio of polychromatic erythrocytes (PCEs) to total erythrocytes (PCEs + NCEs), and the incidence of micronucleated PCEs (MNPCEs/PCEs) were evaluated. The data were analyzed using SPSS Statistics (v21).

## Toxicity to human primary cells

The toxicity of simnotrelvir to human peripheral blood mononuclear cells (PBMCs) (OriCELLs, FPB004) and human primary hepatocytes (BioIVT, F00995-P) was evaluated by the CellTiter-Glo Luminescent Cell Viability Assay (Promega, G7573). PBMCs were cultured in RPMI-1640 medium with 10% FBS, and human primary hepatocytes were cultured in Williams' Medium E with 1% ITS liquid media supplement (Sigma, 13146), 0.1 µM dexamethasone (Sigma, D4902), and 15 µM HEPES (Gibco, 15630-080). For the PBMC assay, the cells were seeded at 54000 cells per well into 96-well plates and incubated overnight. Various concentrations of the compound were added to the cells for 48 h of incubation. The primary hepatocyte assay was carried out by Pharmaron Beijing Co., Ltd. (Beijing, China). In brief, the cells were seeded at 55000 cells per well into collagen I-precoated 96-well plates and incubated for 4–6 h. The supernatant was removed and replaced by fresh medium. After overnight incubation, the supernatant was removed and replaced with fresh medium containing various concentrations of compound. After 48 h of incubation, the medium was replaced every day. Cell viability was determined by the addition of CTG reagent. $CC_{50}$ values were determined using a four-parameter fitting model in GraphPad Prism (v9.1.2) or XLfit software (v5.5.0.5). Experiments were performed in duplicate. Ten concentrations were used to determine the $CC_{50}$ values.

## hERG inhibition assay

The objective of this study was to evaluate the inhibitory effects of simnotrelvir on the hERG current in human embryonic kidney cells stably expressing hERG channels (hERG-HEK293 cells) using a Nanion Patchliner fully automatic patch clamp system (Germany). hERG-HEK293 cells (batch No. S-1-20180605) were obtained from the Li Yang Research Group of Shanghai Institute of Materia Medica, Chinese Academy of Sciences. The cells were clamped using the Nanion Patchliner fully automated patch clamp system to develop a whole-cell voltage clamp mode, and hERG currents were elicited with corresponding voltages. The cells were treated with 0.4 µM cisapride (positive control, batch No. 0000134113, Sigma-Aldrich Co., LLC.), simnotrelvir at 5 concentrations (3, 10, 30, 100, and 300 µM) and DMSO (negative control). Tail currents of hERG channels were recorded to obtain the peak tail current at each concentration using Patchcontrol HT (2.01.26) and Patchmaster (V2×90.3). The data of current were analyzed using Igor Pro software (v6.2.2.2). The rates of hERG current inhibition by 0.4 µM cisapride and by simnotrelvir at different concentrations were calculated with the peak tail current recorded for the negative control set to 100%. Experiments were performed in triplicate.

## Cellular antiviral activity and cytotoxicity evaluation

The African green monkey kidney cell line Vero E6 was purchased from ATCC (ATCC, CRL-1586) and maintained in Dulbecco's modified Eagle's medium (DMEM) with 10% fetal bovine serum (FBS) at 37 °C and 5% $CO_2$. The SARS-CoV-2 isolate WIV04 (GISAID accession number: EPI_ISL_402124) was isolated from Huh7 cells of the original sample and was propagated in Caco-2 cells. The SARS-CoV-2 Delta (B.1.617.2, IVCAS6.7585) and Omicron (B.1.1.529.1, CSTR: 16533.06.IVCAS6.7600) strains were obtained from the National Virus Resource Center and propagated in Vero E6 cells. All experiments using authentic SARS-CoV-2 and its variants were carried out in the Biosafety Level 3 facility of the Wuhan Institute of Virology, Chinese Academy of Sciences.

The cells were seeded at 50000 cells per well into 48-well plates and incubated overnight. A series of concentrations of each compound was added, and the cells were infected with the SARS-CoV-2 WIV04 strain, Delta strain or Omicron strain at a multiplicity of infection (MOI) of 0.01. After inoculation for 1 h, the cell supernatants were removed, and the cells were washed with PBS and treated with fresh medium containing a series of concentrations of compounds. At

24 h (WIV04 and Delta strains) or 48 h (Omicron strain) post-infection, the cell supernatants were collected, and the viral copy numbers were determined by real-time fluorescence quantitative PCR (qRT–PCR). The primers used for the WIV04 and Delta strains were CAATGGTT TAACAGGCACAGG (forward primer) and CTCAAGTGTCTGTGGAT CACG (reverse primer), while CAATGGTTTAAAAGGCACAGG (forward primer) and CTCAAGTGTCTGTGGATCACG (reverse primer) were used for the Omicron strain. All primers were synthesized by Sangon Biotech (China). The $EC_{50}$ values were calculated by GraphPad Prism (v9.1.2). Three independent experiments (each in triplicate) were performed. Six concentrations of each compound were used to calculate the $EC_{50}$ values. The $EC_{50}$ values are expressed as the mean ± SD.

The Vero E6 cell viability assay was evaluated by Cell Counting Kit 8 (CCK8) assay (Beyotime, C0039). Cells were seeded at 20000 cells per well into 96-well plates and incubated overnight. Nine concentrations of each compound were added to the cells and incubated for 24 h. CCK-8 solution (10 μL) was added for 4 h of incubation at 37 °C with 5% $CO_2$. After that, the absorbance at 450 nm was measured using a microplate reader. Three independent experiments (each in triplicate) were performed.

### Animal pharmacokinetic studies

For animal pharmacokinetic studies, animal experiments (dosing and blood sample collection) were carried out at the Shanghai Institute of Materia Medica, Chinese Academy of Sciences for C57BL/6J mice and SD rats and at Kunming Biomed International (KBI) Ltd. for cynomolgus monkeys. The pharmacokinetic studies were approved by the Institute Animal Care and Use Committee of Shanghai Institute of Materia Medica (SIMM) and Kunming Biomed International (KBI) Ltd. Preparation of the standards, analysis of the samples and the subsequent interpretation of results were conducted at Shanghai MEAS Laboratories Co., Ltd. (Shanghai, China). A tissue distribution study of simnotrelvir co-administered with ritonavir to C57BL/6J mice was conducted at Simcere and approved by the Institute Animal Care and Use Committee of Simcere. All procedures related to animal handling, care, and treatment in the pharmacokinetic studies were performed according to approved guidelines.

For the pharmacokinetic study, male C57BL/6J mice (7–8 weeks old, purchased from Shanghai Shengchang Biological Technology Co., Ltd.) were divided into 2 groups ($n = 15$ for each group) and received 50 mg/kg or 200 mg/kg simnotrelvir orally (co-dosed with 50 mg/kg ritonavir) twice daily (BID) for 1 day. After dosing, blood samples were collected from the retro-orbital venous plexus at predetermined timepoints and then centrifuged (12,000 g, 4 °C, 5 min) to separate plasma. SD rats (200-260 g, purchased from Charles River Laboratories (Zhejiang, China)) were fasted for 12 h before dosing and then intravenously or orally dosed with 15 mg/kg simnotrelvir (alone or co-dosed with 30 mg/kg ritonavir) ($n = 3$/sex/group). Blood samples were collected via the retro-orbital venous plexus at predetermined timepoints post dose and then centrifuged to obtain plasma. Cynomolgus monkeys (2-5 kg, 2.5-5 years old, $n = 6$ for each group) were fasted for 12 h before dosing and then intravenously or orally dosed with 5 mg/kg simnotrelvir (alone or orally co-dosed with 15 mg/kg ritonavir) ($n = 3$/sex/group). Blood samples were collected via the limb veins at predetermined timepoints post dose and then centrifuged to obtain plasma. All plasma samples were stored at −70 °C prior to LC-MS/MS analysis. The plasma concentrations of simnotrelvir were analyzed using Phoenix WinNonlin (v7.0).

For the tissue distribution study, male C57BL/6J mice (15−30 g, purchased from Shanghai Lingchang Biotechnology Co., Ltd, China) were orally dosed with 200 mg/kg simnotrelvir (co-dosed with 50 mg/kg ritonavir) ($n = 24$ for each group). After dosing, plasma, lung, and brain samples were collected at predetermined timepoints and subjected to LC-MS/MS analysis. To reduce the contamination from

the blood, we bled the mice adequately. After that, the brain was washed by the physiological saline.

### Repeat dose toxicity studies

The toxicology study design and parameters evaluated were in compliance with the US FDA, OECD and NMPA GLP regulations. All procedures related to animal handling, care and treatment in repeat-dose toxicity studies were performed according to approved guidelines. SD rats were randomly divided into five groups ($n = 15$/sex/group) and orally administered vehicle (98% (v/v) 0.5% (w/v) methyl cellulose solution with 2% (v/v) Tween 80), control formulation (1.8% MTBE in vehicle), or simnotrelvir (30, 180, or 1000 mg/kg) once daily for 14 consecutive days followed by a 14-day recovery period. Parameters including clinical observations, ophthalmology, body weight, food consumption, hematology, coagulation, plasma chemistry, urinalysis, gross pathology, organ weights, and histopathology were evaluated. A concurrent toxicokinetic evaluation was also performed. SD rats were randomly divided into 4 groups ($n = 4$/sex/group). Cynomolgus monkeys were orally administered the control formulation (1.1% MTBE in vehicle, n = 5/sex/group) or simnotrelvir in vehicle at doses of 40 ($n = 3$/sex/group), 160 ($n = 3$/sex/group) and 600 mg/kg/day ($n = 5$/sex/group) twice daily for 14 consecutive days followed by a 14-day recovery period. Parameters including clinical observations, ophthalmology, body weight, food consumption, body (rectal) temperature, clinical pathology (hematology, coagulation, plasma chemistry, and urinalysis), pathology (gross pathology, organ weight and histopathology) were recorded and analyzed using Provantis (v10.2.3.1). Toxicokinetics parameters were collected using LCMS system Analyst software (v1.7) and Watson LIMS (v7.5 SP1), and analyzed using WinNonlin (v6.3). Cardiovascular system safety pharmacology parameters (HR, PR interval, QRS duration, QT interval, QTcF interval, and blood pressure) were collected using Iox2 (v10.0.40) and analyzed using ecgAUTO (v3.3.5.10).

### In vivo efficacy study

The in vivo efficacy study was performed in an animal biosafety level 3 facility at the Laboratory Animal Center of Wuhan Institute of Virology, Chinese Academy of Sciences. Protocols were approved by the Institutional Animal Care and Use Committee at the Wuhan Institute of Virology. All procedures related to animal handling, care and treatment in efficacy studies were performed according to approved guidelines. The mice were housed in transparent plastic cages placed on stainless steel rack in standard SPF animal rooms with a 12-h light/12-h dark cycle, a constant temperature from 20 to 26 °C, and a humidity from 40% to 70%. K18-hACE2 mice (GemPharmatech, 7 to 8 weeks old, male) were divided into 5 groups: Group 1, vehicle (untreated, infected control, $n = 10$); Group 2, nirmatrelvir 200 mg/kg (co-dosed with 50 mg/kg ritonavir, n = 10); Group 3, simnotrelvir 200 mg/kg (co-dosed with 50 mg/kg ritonavir, $n = 10$); Group 4, simnotrelvir 50 mg/kg (co-dosed with 50 mg/kg ritonavir, $n = 10$); and Group 5, mock infection (untreated, uninfected control, $n = 2$). Two mice, one from group 1 and the other from group 4, were not included for data collection due to the damage caused by improper intragastric administration. The SARS-CoV-2 Delta (B.1.617.2, IVCAS6.7585) strain was obtained from the National Virus Resource Center and propagated in Vero E6 cells. The mice were anesthetized by intraperitoneal injection of 2.5% avertin (20 μL/g body weight) and inoculated intranasally with $1×10^3$ PFU SARS-CoV-2 Delta strain at Day 0. Simnotrelvir and ritonavir were solubilized in 85% saline with 5% DMSO, 5% Solutol HS 15 and 5% PEG 400. Mice were dosed BID for 2 days or 4 days beginning at 2 h post-infection. The weights of the mice were monitored daily from Day 0 to the end of the study. Half of the mice in each group except for those in the mock infection group were euthanized by isoflurane inhalation on Day 2, and the others were euthanized on Day 4. Lung and brain tissues were collected for viral load detection, immunohistochemistry and histopathology assessments. For viral copy

number detection, tissues were homogenized in DMEM, and RNA was extracted by an RNeasy Mini Kit (Qiagen, #74104) for real-time quantitative PCR (RT–PCR). Viral titers were determined by plaque assay following a general procedure. For tissue histopathology assessment, the tissues were fixed in 4% formaldehyde, dehydrated, embedded, sectioned, and stained with H&E for histopathology assessment. Details on the scoring systems are summarized in Supplementary Tables 5 and 6. For lung tissue samples, four parameters were evaluated: alveolar atrophy or dilation, alveolar wall thickening, alveolar hemorrhage, and infiltration of inflammatory cells. For brain tissues, three parameters were evaluated: bleeding and neuron degeneration in the hippocampus and cortex. For the tissue immunohistochemistry assessment, tissue sections were blocked with blocking buffer to reduce background signal and incubated with anti-SARS-CoV-2 nucleocapsid protein monoclonal antibody (Cell Signaling Technology, #26369) at a 1:400 dilution and then with secondary polyclonal antibody (SeraCare, #5220-0336) at a 1:500 dilution. The sections were viewed and scanned by digital microscopy. Statistical significance was evaluated by using GraphPad Prism (v9.1.2). Changes in viral load and histopathology were compared to the vehicle group and analyzed by one-way ANOVA or two tailed Welch's $t$ test.

### Reporting summary

Further information on research design is available in the Nature Portfolio Reporting Summary linked to this article.

## Data availability

The atomic coordinates and structure factors have been deposited into the Protein Data Bank with accession codes 8IFP (SARS-CoV-2 3 CL^pro in complex with compound **1**), 8IFQ (SARS-CoV-2 3CL^pro in complex with compound **2**), 8IFR (SARS-CoV-2 3CL^pro in complex with compound **3**), 8IFS (SARS-CoV-2 3CL^pro in complex with compound **7**), 8IFT (SARS-CoV-2 3CL^pro in complex with compound **10**), 8IGX (SARS-CoV-2 3CL^pro in complex with simnotrelvir), and 8IGY (SARS-CoV-2 3CL^pro in complex with nirmatrelvir). The structure of SARS-CoV-2 3CL^pro in complex with boceprevir (PDB code: 6XQU) was obtained from Protein Data Bank. The cDNA of 3CL^pro's of SARS-CoV-2 (Gen-Bank: MN908947.3), SARS-CoV (Gen-Bank: AAP13442.1), MERS-CoV (Gen-Bank: MT387202.1), H229E-CoV (Gen-Bank: AF304460.1), HKU1-CoV (Gen-Bank: AY597011.2), NL63-CoV (Gen-Bank: AY567487.2), and OC43-CoV (Gen-Bank: AY903459.1) were obtained from Genbank [https://https.ncbi.nlm.nih.gov/genbank/]. Source data are provided with this paper. All the raw data generated in this study are provided in the Source Data file. Source data are provided with this paper.

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

## Acknowledgements

The authors thank the staff from beamlines BL02U1, BL17U1, BL18U1, and BL19U1 at Shanghai Synchrotron Radiation Facility for assistance during data collection. We also thank Hongping Wei, Tao Du, Jin Xiong, and Min Hou from the BSL-3 Laboratory. This work was supported by the National Natural Science Foundation of China (No. 22277130 to Y.X. and No. U22A20379 to G.X), the Shanghai Science and Technology Committee grant (No. 22YF1457300) to H.S., the Qiusuo Outstanding Youth Project of Lingang Laboratory (No. LG-QS-202205-02) to H.S., the Shanghai Institute of Materia Medica, CAS (No. CASIMM0120234003 to J.S., No. SIMM010120 to X.J., and No. SIMM010118 to Y.X.), and the Wuhan Institute of Virology, CAS (2022QNTJ-01) to L.Z.. This paper is dedicated to the memory of professor Hualiang Jiang, for his scientific leadership and impact on the discovery of simnotrelvir.

## Author contributions

Y.X., X.J., and J.S. conceived and design the project. Y.X., X.J., J.S., H.J., L.Z., H.S., F.Z., and R.T. designed the experiments; Y.X., X.J., J.S., H.S., and M.X. performed the drug design; X.J., J.S., Y.Z., Q.Z., and F.Y. performed the chemical experiments and collected the data; H.S., W.Z., and M.L. performed protein expression and purification, crystallization, X-ray diffraction data collection, solved and analyzed the crystal structures; H.S. H.X. and T.N. performed enzymatic assays and determined the $IC_{50}$ values of the compounds; W.S., L.Z., and G.X. performed antiviral cellular assays, cellular cytotoxicity assay and in vivo antiviral studies; F. Z., L.J., X.H., P.C., S.P., and R.T. performed the PK, toxicity and safety studies; Y.X. and H.S. wrote and revised the manuscript with input from all other authors.

## Competing interests

X.J., Y.X., L.Z., H.S., Q.Z., W.Z., W.S., J.S., G.X., and H.J. are co-inventors of the patent (CN202111168232.4, China) that cover 3CL[pro] inhibitors included in this study. F. Z., L.J., X.H., P.C., S.P., and R.T. are employers of Simcere Co. Ltd. F. Z., L.J., X.H., P.C., S.P., and R.T. are shareholders in Simcere Co. Ltd. All other authors declare no competing interests.

## Additional information

[1]State Key Laboratory of Drug Research, Shanghai Institute of Materia Medica, Chinese Academy of Sciences, 201203 Shanghai, China. [2]University of Chinese Academy of Sciences, 100049 Beijing, China. [3]State Key Laboratory of Virology, Wuhan Institute of Virology, Center for Biosafety Mega-Science, Chinese Academy of Sciences, 430071 Wuhan, China. [4]State Key Laboratory of Neurology and Oncology Drug Development, 210023 Nanjing, China. [5]Simcere Zaiming Pharmaceutical Co., Ltd, 200000 Shanghai, China. [6]School of Chinese Materia Medica, Nanjing University of Chinese Medicine, 210023 Nanjing, China. [7]Shanghai Synchrotron Radiation Facility, Shanghai Advanced Research Institute, Chinese Academy of Sciences, 201210 Shanghai, China. [8]Jiangsu Simcere Pharmaceutical Co., Ltd, 210023 Nanjing, China. [9]Hubei jiangxia Laboratory, 430200 Wuhan, China. [10]School of Pharmaceutical Science and Technology, Hangzhou Institute for Advanced Study, University of Chinese Academy of Sciences, 310024 Hangzhou, China. [11]These authors contributed equally: Xiangrui Jiang, Haixia Su, Weijuan Shang, Feng Zhou, Yan Zhang, Wenfeng Zhao. [12]Deceased: Hualiang Jiang. ✉e-mail: renhong.tang@simceregroup.com; zhangleike@wh.iov.cn; shenjingshan@simm.ac.cn; ycxu@simm.ac.cn

