## [Peer Review File · Nature Communications]

Structure-based development and preclinical evaluation of the SARS-CoV-2 3C-like protease inhibitor simnotrelvirREVIEWER COMMENTS

Reviewer #1 (Remarks to the Author):

Xu et al report the structure-based design and discovery of simnotrelvir, an orally active SARS-CoV-2 Mpro inhibitor, and its preclinical data. Simnotrelvir shows excellent activity against Mpro, high cellular antiviral activity, good pharmacokinetic and safety profiles, and robust oral efficacy in a transgenic mouse model of SARS-CoV-2 Delta infection. This compound has received conditional approval from NMPA of China. The manuscript is well organized and written. Overall, it deserves to be published in Nature Communications.

Several minor revisions:

1. The number of decimal places should be unified. I suggest to keep at most one decimal place for nanomolar, and at most three decimal places for micromolar.
2. Some images have lower resolution (e.g. 4). Pls provide higher resolution images in the manuscript.

Reviewer #2 (Remarks to the Author):

In their manuscript NCOMMS-23-16705-T, Jiang et al describe the discovery and the biological profile of simnotrelvir, a peptidomimetic inhibitor of the main protease (Mpro) of SARS-CoV-2. Simnotrelvir is in Phase 2/3 clinical development and received conditional approval in China, and this paper is the first disclosure of the drug's properties. The drug was found by re-designing the HCV protease inhibitor boceprevir to improve its Mpro binding properties. The medicinal chemistry optimization is described concisely and is a good example for structure- and biophysics-based drug design. The use of a C145G mutant that removes covalent binding is an elegant solution to study binding enthalpy. The compounds are highly pure, and the synthesis and analytical characterization is fully documented. Also the biological characterization, comprising the demonstration of high in vitro potency and selectivity, pharmacokinetic data and in vivo efficacy in relevant mouse models of SARS-CoV-2 infection, is sound and convincing. Overall, I have mostly minor suggestions for technical improvements.

The major weakness of the paper concerns the degree of novelty over prior art. The structure-based boceprevir redesign strategy has been published by several groups before (see ref 20, Göhl et al 2022, etc), and key improvements such as the gamma-lactam introduction are well-known. Simnotrelvir is a very close analog of Pfizer's nirmatrelvir, with the dimethylcyclopropyl moiety replaced by a dithiazaspiro-group. It is a classical 'me-too' drug, although found independently. Main limitations of nirmatrelvir, such as the need to combine with ritonavir, are not overcome with simnotrelvir, and the overall profile of the drugs are similar, in spite of subtle differences in the binding mode, such as the interesting S-S contacts observed with simnotrelvir.

Minor comments

- Figure 1: I suggest using the same units for different compounds and different parameters, ideally nM. For example, Kd is expressed in nM, μ M, and M in the current version.
- Figure 2: I think the color of S1 is magenta, and that of S1' is missing
- Figure 3/safely profile: The most relevant class of off-targets are cathepsins due to their close similarity to Mpro. The authors report only inactivity against cathepsin D. I suggest testing a larger panel of cathepsins, because they are pharmacologically relevant.
- Ref 4 is redundant, given reference 5
- Many expressions and sentences in the manuscript are not fully clear, past and present tense switch often, etc. The text is understandable, but it can be improved. A sentence-by-sentence editing is required before publication.
- The result section ends with '...., providing an additional contribution to efficacy in the brains.' This statement is not supported by data, because efficacy in the brain has not been shown.
- Determining brain concentrations is technically difficult, because contaminations with blood are hard to avoid. The authors did not comment on this, but the fact that at later time points, brain concentrations exceed those in the plasma give me confidence in the data.

Reviewer #3 (Remarks to the Author):

Jiang et al report the design and testing of an inhibitor towards the 3CL-like protease (3CLPro, also known as Main protease, MPro) encoded by SARS-CoV-2. The compound simnotrelvir, is a covalent inhibitor, which reacts with the active site residue Cys145 in 3CLPro. This drug was conditionally approved in January 2023 for treatment of mild to moderate Covid-19 by the National Medical Products Administration in China.

The authors combine a range of techniques to provide a comprehensive characterization of the properties of simnotrelvir and structure-activity relationships with related compounds: from crystal structures, enzymatic and binding assays, cellular and safety profile assays to in vivo antiviral testing in animal models. The structural and biochemical characterization of simnotrelvir and related compounds are largely of high quality.

Obviously, conceptually (and structurally) simnotrelvir is very similar to Pfizer's drug Nirmatrelvir (included in Paxlovid). But I leave any decisions about novelty and impact to the editors.

I wonder about the enzymatic assay to determine IC50 values. According to Materials, 40 nM of SARS-CoV-2 3CLPro was used. But several of the reported IC50 values in Figure 1 are lower than this enzyme concentration, so how should these IC50 values be interpreted? These experiments might need to be repeated with an excess inhibitor. (Lower enzyme concentration and longer reaction times can be used.) Furthermore, Simnotrelvir forms a covalent bond, which can reversibly break and be in equilibrium with the enzyme. Did you check that the kinetics of inhibitor binding (ie, time to reach equilibrium) is much faster than the steady state rates (likely yes)? However, what is the meaning of IC50 values for irreversible inhibitors such as compound 3, the values must only be dependent on the incubation time with enzyme and on the stoichiometry?

Thermodynamics, Discussion (and which I propose should be moved to results, see below); As the authors point out, the relationship between enthalpy and entropy and how they contribute to binding is complex. Indeed, both ΔH and ΔS are usually very temperature dependent, and deeper insight therefore requires determination of ΔC_p ($=d\Delta H/dT$), which correlates with burial of hydrophobic surface upon binding. The authors highlight a difference between Simnotrelvir and Nirmatrelvir with regard to ΔH with more beneficial (enthalpic) interactions made by Simnotrelvir (Figure 2). For this to be convincing I would suggest measuring a temperature dependence of ΔH to get ΔC_p . Or at least perform the experiments at body temperature (I could not find the experimental temperature in the manuscript).

Minor revisions:

Give experimental temperatures for all enzymatic and binding experiments.

Revise the entire manuscript with regard to the number of significant digits in the reported parameters (round off). For example, on page 15, IC50 values of 28.26, 25.42 and 43.91 nM are reported, whereas 28, 25 and 44 nM are more reasonable. That is, use two significant numbers for IC50, Kd values and half-lives. This also applies to all figures where parameters are shown.

Large parts of the Discussion would fit better in the Results section, in particular most of p. 15 and the upper part of p. 17. The Discussion can then be more focused on comparisons with other 3CLPro/protease inhibitors.

We want to thank the reviewers for positive comments, constructive suggestions and thorough review of this work. We have revised our manuscript to fully address all comments and suggestions made by three reviewers. Below are our point-by-point responses (colored *blue*) to the Reviewers' comments (colored *black*). For your convenience, all the changes in the text made in response to the comments have been highlighted in *red* in the revised manuscript.

Responses to Reviewer 1:

Comments: Xu et al report the structure-based design and discovery of simnotrelvir, an orally active SARS-CoV-2 Mpro inhibitor, and its preclinical data. Simnotrelvir shows excellent activity against Mpro, high cellular antiviral activity, good pharmacokinetic and safety profiles, and robust oral efficacy in a transgenic mouse model of SARS-CoV-2 Delta infection. This compound has received conditional approval from NMPA of China. The manuscript is well organized and written. Overall, it deserves to be published in Nature Communications.

Response: We thank the reviewer for the positive comments and finding our work appropriate for publication in *Nature Communications*.

Comments: The number of decimal places should be unified. I suggest to keep at most one decimal place for nanomolar, and at most three decimal places for micromolar.

Response: Thank the reviewer for this comment. As three reviewers all made the comment on unification of decimal places and units, we used the same unit 'nM' for IC_{50} , K_d , $GSH_{t1/2}$, and EC_{50} and keep their values as integers in the revised manuscript.

Comments: Some images have lower resolution (e.g. 4). Pls provide higher resolution images in the manuscript.

Response: Thank the reviewer for this comment. The low-resolution images were generated when the website converted Word documents into PDF. During this revision submission, we will upload each figure with a resolution of 600 dpi as a separate file to avoid this problem.

Responses to Reviewer 2:

Comments: In their manuscript NCOMMS-23-16705-T, Jiang et al describe the discovery and the biological profile of simnotrelvir, a peptidomimetic inhibitor of the main protease (M^{pro}) of SARS-CoV-2. Simnotrelvir is in Phase 2/3 clinical development and received conditional approval in China, and this paper is the first disclosure of the drug's properties. The drug was found by re-designing the HCV protease inhibitor boceprevir to improve its M^{pro} binding properties. The medicinal chemistry optimization is described concisely and is a good example for structure- and biophysics-based drug design. The use of a C145G mutant that removes covalent binding is an elegant solution to study binding enthalpy. The compounds are highly pure, and the synthesis and analytical characterization is fully documented. Also the biological characterization, comprising the demonstration of high in vitro potency and selectivity, pharmacokinetic data and in vivo efficacy in relevant mouse models of SARS-CoV-2 infection, is sound and convincing. Overall, I have mostly minor suggestions for technical improvements.

Response: We thank the reviewer for thoroughly reading our manuscript and thinking that the discovery of simnotrelvir is a good example for structure- and biophysics-based drug design. We are also grateful to the reviewer for all these positive comments on our work.

Comments: The major weakness of the paper concerns the degree of novelty over prior art. The structure-based boceprevir redesign strategy has been

published by several groups before (see ref 20, Göhl et al 2022, etc), and key improvements such as the gamma-lactam introduction are well-known. Simnotrelvir is a very close analog of Pfizer's nirmatrelvir, with the dimethylcyclopropyl moiety replaced by a dithiazaspiro-group. It is a classical 'me-too' drug, although found independently. Main limitations of nirmatrelvir, such as the need to combine with ritonavir, are not overcome with simnotrelvir, and the overall profile of the drugs are similar, in spite of subtle differences in the binding mode, such as the interesting S-S contacts observed with simnotrelvir.

Response: We agree with the reviewer that the structure-based boceprevir redesign strategy has been published previously and the chemical structure of simnotrelvir is very close to that of Pfizer's nirmatrelvir. However, simnotrelvir was developed independently, and it is also the first peptidomimetic inhibitor derived from boceprevir that entered the phase I-III clinical trials and received the conditional approval. Most importantly, the dithiazaspiro-proline group which is distinct to the dimethylcyclopropyl-proline of boceprevir and nirmatrelvir was used as a building block of an peptidomimetic inhibitor for the first time. We have checked the complex structures available in Protein Data Bank (PDB), and found that only three types (I-III) of cyclic segments were used in 161 different peptidomimetic inhibitors of SARS-CoV-2 3CL^{pro} and the main segment is the dimethylcyclopropyl-proline (Supplementary Fig. 14). Furthermore, we also analyzed the interactions between these cyclic segments and SARS-CoV-2 3CL^{pro}. The dithiazaspiro-proline group, which has rarely been exploited for drug design, has been proved to be fit well into the S2 subsite of SARS-CoV-2 3CL^{pro}. It showed that more hydrophobic interactions have been formed between the dithiazaspiro-proline group and S2 subsite than other cyclic segments (Supplementary Fig. 14). Therefore, the dithiazaspiro-proline group provided a good candidate for design of oral peptidomimetic inhibitors targeting 3CL^{pro}s and other proteases. We have added the new figure

(Supplementary Fig. 14) in the revised Supplementary Information and related text in Discussion in the revised manuscript (Lines 519-533, Page 18-19).

In addition, simnotrelvir plus ritonavir have proved to be effective in a phase II-III clinical study. The manuscript reporting the clinical research results was submitted the New England Journal of Medicine (NEJM) and a revised version of the manuscript was just sent back to NEJM (23-01425.R1). The phase II-III clinical trial studies have shown that early administration of simnotrelvir plus ritonavir demonstrates strong antiviral effects, is effective in shortening time to symptom recovery among adult COVID-19 patients, and has an acceptable safety profile. This also indicated that the dithiazaspiro-proline could serve as a novel cyclic segment for the design of oral protease inhibitors.

Notably, in the present study, the structure-activity relationships of a series of ligands toward the SARS-CoV-2 3CL^{pro} are comprehensively understood based on an integration of the ligand binding enthalpy/entropy data and the protein–ligand interaction pattern precisely determined by high-quality crystal structures, which is thus critical to efficiently guide rational drug design and eventually lead to simnotrelvir with optimal potency. Therefore, the simnotrelvir discovery process shows the elegance and utility of structure-based drug design and provides a good example for structure- and biophysics-based drug design.

Comments: Figure 1: I suggest using the same units for different compounds and different parameters, ideally nM. For example, K_d is expressed in nM, μM, and M in the current version.

Response: Following the three reviewers' comment on unification of units, we used nM for different compounds and different parameters in the revised manuscript.

Comments: Figure 2: I think the color of S1 is magenta, and that of S1' is missing.

Response: we sincerely thank the reviewer for this comment. We have corrected it in the revised manuscript (Line 604, Page 23). The colors of S1' and S1 are cyan and magenta respectively.

Comments: Figure 3/safely profile: The most relevant class of off-targets are cathepsins due to their close similarity to Mpro. The authors report only inactivity against cathepsin D. I suggest testing a larger panel of cathepsins, because they are pharmacologically relevant.

Response: According to the reviewer's suggestion, we have measured the selectivity of simnotrelvir against cathepsins A/B/D/E/L. The results showed that simnotrelvir demonstrated no significant inhibitory activities against cathepsins A/D/E/L at a concentration of 100 μ M. A weak inhibitory effect was observed with simnotrelvir on cathepsin B (IC_{50} : 7194 nM), but the resulted selectivity index (SI) was 799. It indicates that simnotrelvir showed satisfied selectivity against human cathepsins and have a good *in vitro* safety profile. These results have been added in the revised manuscript (Lines 329-333, Page 12). In addition, the pre-clinical as well as the phase I-III clinical trials studies have shown that simnotrelvir has an acceptable *in vivo* safety profile.

Comments: Ref 4 is redundant, given reference 5.

Response: Following the reviewer's suggestion, Ref 4 has been deleted in the revised manuscript (Page 42).

Comments: Many expressions and sentences in the manuscript are not fully clear, past and present tense switch often, etc. The text is understandable, but it can be improved. A sentence-by-sentence editing is required before publication.

Response: Following the reviewer's suggestion, we sent our manuscript to Nature Research for English Language Editing. The certification as well as the record of the editing was provided as a separate file, named "Certification and

record of the editing". With the help of such an editing service, the grammar, spelling mistakes, and other language mistakes in the manuscript have been fixed. In addition, a thorough proofreading and a sentence-by-sentence editing have also been done by the authors.

Comments: The result section ends with '...., providing an additional contribution to efficacy in the brains.' This statement is not supported by data, because efficacy in the brain has not been shown.

Response: We have deleted the sentence 'providing an additional contribution to efficacy in the brains.' in the revised manuscript (Line 431, Page 15).

Comments: Determining brain concentrations is technically difficult, because contaminations with blood are hard to avoid. The authors did not comment on this, but the fact that at later time points, brain concentrations exceed those in the plasma give me confidence in the data.

Response: We agree with the reviewer that determining brain concentrations is technically difficult. So we have paid attention to the contaminations with the blood in the process of the experiments. To reduce the contaminations with blood, we firstly bleed the mice adequately. After that, we wash the brain with the physiological saline. We also agree with the reviewer that contaminations with blood are hard to avoid after these procedures. As the reviewer said, brain concentrations exceed those in the plasma at 24 h. In this context, the blood has little influence on the determination of brain concentrations. This data convinced us that simnotrelvir could be distributed to the brain. Following the reviewer's suggestion, we have added the description of the procedure in the method (Lines 983-984, Page 40) and also mention it in the main text (Lines 423-426, Page15).

Responses to Reviewer 3:

Comments: Jiang et al report the design and testing of an inhibitor towards the 3CL-like protease (3CLPro, also known as Main protease, MPro) encoded by SARS-CoV-2. The compound simnotrelvir, is a covalent inhibitor, which reacts with the active site residue Cys145 in 3CLPro. This drug was conditionally approved in January 2023 for treatment of mild to moderate Covid-19 by the National Medical Products Administration in China.

The authors combine a range of techniques to provide a comprehensive characterization of the properties of simnotrelvir and structure-activity relationships with related compounds: from crystal structures, enzymatic and binding assays, cellular and safety profile assays to in vivo antiviral testing in animal models. The structural and biochemical characterization of simnotrelvir and related compounds are largely of high quality.

Response: We are grateful for the positive comments given by the reviewer.

Comments: Obviously, conceptually (and structurally) simnotrelvir is very similar to Pfizer's drug Nirmatrelvir (included in Paxlovid). But I leave any decisions about novelty and impact to the editors.

Response: As the response to the second reviewer, we agree with the reviewer that the chemical structure of simnotrelvir is very close to that of Pfizer's nirmatrelvir. However, simnotrelvir was developed independently, and it is also the first peptidomimetic inhibitor derived from boceprevir that entered the phase I-III clinical trials and received the conditional approval. Most importantly, the dithiazaspiro-proline group which is distinct to the dimethylcyclopropyl-proline of boceprevir and nirmatrelvir was used as a building block of an peptidomimetic inhibitor for the first time. We have checked the complex structures available in Protein Data Bank (PDB), and found that only three types (I-III) of cyclic segments were used in 161 different peptidomimetic inhibitors of SARS-CoV-2 3CL^{Pro} and the main segment is the dimethylcyclopropyl-proline (Supplementary Fig. 14). Furthermore, we also analyzed the interactions between these cyclic segments and SARS-CoV-2 3CL^{Pro}. The dithiazaspiro-

proline group, which has rarely been exploited for drug design, has been proved to be fit well into the S2 subsite of SARS-CoV-2 3CL^{pro}. It showed that more hydrophobic interactions have been formed between the dithiazaspiro-proline group and S2 subsite than other cyclic segments (Supplementary Fig. 14). Therefore, the dithiazaspiro-proline group provided a good candidate for design of oral peptidomimetic inhibitors targeting 3CL^{pro}s and other proteases. We have added the new figure (Supplementary Fig. 14) in the revised Supplementary Information and related text in Discussion in the revised manuscript (Lines 519-533, Page 18-19).

In addition, simnotrelvir plus ritonavir have proved to be effective in a phase II-III clinical study. The manuscript reporting the clinical research results was submitted the New England Journal of Medicine (NEJM) and a revised version of the manuscript was just sent back to NEJM (23-01425.R1). The phase II-III clinical trial studies have shown that early administration of simnotrelvir plus ritonavir demonstrates strong antiviral effects, is effective in shortening time to symptom recovery among adult COVID-19 patients, and has an acceptable safety profile. This also indicated that the dithiazaspiro-proline could serve as a novel cyclic segment for the design of oral protease inhibitors.

Notably, in the present study, the structure-activity relationships of a series of ligands toward the SARS-CoV-2 3CL^{pro} are comprehensively understood based on an integration of the ligand binding enthalpy/entropy data and the protein–ligand interaction pattern precisely determined by high-quality crystal structures, which is thus critical to efficiently guide rational drug design and eventually lead to simnotrelvir with optimal potency. Therefore, the simnotrelvir discovery process shows the elegance and utility of structure-based drug design and provides a good example for structure- and biophysics-based drug design.

Comments: I wonder about the enzymatic assay to determine IC₅₀ values. According to Materials, 40 nM of SARS-CoV-2 3CL^{Pro} was used. But several

of the reported IC₅₀ values in Figure 1 are lower than this enzyme concentration, so how should these IC₅₀ values be interpreted? These experiments might need to be repeated with an excess inhibitor. (Lower enzyme concentration and longer reaction times can be used.)

Response: Following the reviewer's suggestion, we used lower enzyme concentration and longer reaction times to re-measured all IC₅₀ values of compounds against SARS-CoV-2 3CL^{pro} and its mutants. To this end, we firstly tried various concentrations of SARS-CoV-2 3CL^{pro} (40 nM, 20 nM, 10 nM, 5 nM, and 2.5 nM) plus substrate and calculated the Z factors between the negative and positive controls to assess the quality of the screening assays. Z factors were calculated to be 0.90, 0.91, 0.87, 0.68, and 0.37 for the protease concentration of 40 nM, 20 nM, 10 nM, 5 nM, and 2.5 nM, respectively (Fig. R1 shown below). As the Z-factor value decreased sharply to 0.68 at 5 nM, we used 10 nM as the final concentration of the protease. The reaction rate of 10 nM recombinant SARS-CoV-2 3CL^{pro} hydrolyzing 10 μM fluorogenic substrates is linear within 1 h, so 1 h was used as the reaction time. We have re-measured all IC₅₀ values with the protein concentration of 10 nM and the reaction time of 1 h. The new measured IC₅₀ values have been used to replace the old ones in the revised manuscript.

Fig. R1. The calculated Z factors between the positive controls added with various concentrations of SARS-CoV-2 3CL^{pro} and negative controls.

Comments: Furthermore, Simnotrelvir forms a covalent bond, which can reversibly break and be in equilibrium with the enzyme. Did you check that the kinetics of inhibitor binding (ie, time to reach equilibrium) is much faster than the steady state rates (likely yes)? However, what is the meaning of IC₅₀ values for irreversible inhibitors such as compound 3, the values must only be dependent on the incubation time with enzyme and on the stoichiometry?

Response: We agree with the reviewer that the kinetics of simnotrelvir noncovalently binding to 3CL^{pro} should be faster than the steady state rates, as the inhibitor binding is only one of events (inhibitor binding, covalent bond form and break) occurred on the process to reach the steady-state inhibition of the protease. We have performed ITC experiments to roughly address this point. The results showed that the time required for simnotrelvir reaching the steady state inhibition of SARS-CoV-2 3CL^{pro} is longer than the time required for simnotrelvir noncovalently binding to SARS-CoV-2 3CL^{pro} C145G mutation (Fig. R2 shown below).

Fig. R2. The heat change over time after simnotrelvir injecting into SARS-CoV-2 3CL^{pro} C145G mutation (a) and wild type (b).

We also agree with the reviewer that different incubation time used to measure the IC₅₀ of an irreversible inhibitor will result in different IC₅₀ values. We measured the IC₅₀ values of compound 3 against SARS-CoV-2 3CL^{pro} after incubating for various times (0, 5, 10, 15, and 20 min). The results showed that

the IC_{50} value of compound 3 against SARS-CoV-2 3CL^{pro} was dependent on the incubation time (Fig. R3 shown below). However, measurement of IC_{50} values is economically efficient, and it helps us to rapidly rule out some irreversible inhibitors with poor potency if the same incubation time is used for all the measurements. In our manuscript, the incubation time used for the IC_{50} values measurement was all provided in the method. For the irreversible inhibitors with satisfied IC_{50} values, other parameters, such as non-covalent affinity (K_i), the reaction rate of the bound inhibitor (K_{inact}), and inherent reactivity towards nonspecific thiols, should be further evaluated (*Nat. Rev. Drug Discov.* 2011, 10(4):307-17). An ideal irreversible inhibitor would share potent non-covalent affinity, fast reaction rate of the bound inhibitor and low inherent reactivity. In the present manuscript, we found the IC_{50} value of compound 3 against SARS-CoV-2 3CL^{pro} is acceptable compared to the other two inhibitors. Therefore, we next evaluated the inherent reactivity of compound 3 by determining the $GSH_{t1/2}$. However, the data showed that the inherent reactivity of compound 3 is not satisfied. As a result, we did not select compound 3 for further optimization.

Fig. R3. Concentration-dependent inhibition of SARS-CoV-2 3CL^{pro} by compound 3 after incubation of 0, 5, 10, 15, and 20 min, respectively.

Comments: Thermodynamics, Discussion (and which I propose should be moved to results, see below); As the authors point out, the relationship between

enthalpy and entropy and how they contribute to binding is complex. Indeed, both ΔH and $T\Delta S$ are usually very temperature dependent, and deeper insight therefore requires determination of ΔC_p ($=d\Delta H/dT$), which correlates with burial of hydrophobic surface upon binding. The authors highlight a difference between Simnotrelvir and Nirmatrelvir with regard to ΔH with more beneficial (enthalpic) interactions made by Simnotrelvir (Figure 2). For this to be convincing I would suggest measuring a temperature dependence of ΔH to get ΔC_p . Or at least perform the experiments at body temperature (I could not find the experimental temperature in the manuscript).

Response: We sincerely thank the reviewer for this constructive suggestion. The initial ITC experiments in the manuscript were performed at 30 °C, which has been added in the revised manuscript (Line 776, Page 33). To determine the ΔC_p of SARS-CoV-2 3CL^{pro} binding with simnotrelvir and nirmatrelvir, we have also performed the ITC experiments at 20 °C and 37 °C (body temperature suggested by the reviewer), respectively. The results showed that the ΔC_p ($-0.63 \text{ kJ mol}^{-1} \text{ K}^{-1}$) to form the SARS-CoV-2 3CL^{pro}–simnotrelvir complex was more negative than that ($-0.41 \text{ kJ mol}^{-1} \text{ K}^{-1}$) of the SARS-CoV-2 3CL^{pro}–nirmatrelvir complex (Fig. R4 shown below). It is thus indicated that, compared to nirmatrelvir, binding of simnotrelvir to SARS-CoV-2 3CL^{pro} leads to burial of a greater hydrophobic surface area, which is in accordance with the result revealed by crystal structures. In addition, the ITC experiments performed at 37 °C also revealed that enthalpy of simnotrelvir ($\Delta H = -39.68 \pm 0.61 \text{ kJ/mol}$) binding to SARS-CoV-2 3CL^{pro} is more significant than that of nirmatrelvir ($\Delta H = -29.23 \pm 1.34 \text{ kJ/mol}$) at body temperature. These results have been added in the revised manuscript (Lines 276-285, Page 10 in the revised manuscript, and Supplementary Fig. 10 in the revised Supplementary Information).

Fig. R4. Temperature dependence of the measured enthalpy (ΔH) of SARS-CoV-2 3CL^{pro} binding to nirmatrelvir (red) and simnotrelvir (blue). Three independent experiments were performed at each temperature.

Comments: Give experimental temperatures for all enzymatic and binding experiments.

Response: Following the reviewer's suggestion, the experimental temperatures for the enzymatic and binding experiments have been added in the revised manuscript (Lines 717, 735, 747, 759, 776, 779, 800, 812, and 824).

Comments: Revise the entire manuscript with regard to the number of significant digits in the reported parameters (round off). For example, on page 15, IC₅₀ values of 28.26, 25.42 and 43.91 nM are reported, whereas 28, 25 and 44 nM are more reasonable. That is, use two significant numbers for IC₅₀, K_d values and half-lives. This also applies to all figures where parameters are shown.

Response: Following the three reviewers' comment on unification of decimal places and units, we used the unit 'nM' for IC₅₀, K_d, GSH_{t1/2} and EC₅₀ and keep the values as integers in the revised manuscript.

Comments: Large parts of the Discussion would fit better in the Results section, in particular most of p.15 and the upper part of p.17. The Discussion can then be more focused on comparisons with other 3CLPro/protease inhibitors.

Response: Following the reviewer's suggestion, we have moved most text in p.15 and the upper part of p.17 to Result section (Pages 8-11, and 16-20). In addition, we have added the content regarding the comparisons of simnotrelvir with 161 peptidomimetic inhibitors and another approved SARS-CoV-2 3CL^{pro} inhibitor (ensitrelvir) in the revised Discussion section (Pages 18-19).

With these changes we hope we have addressed all comments. We would like to thank the reviewers again for the constructive, professional and helpful suggestions.

REVIEWERS' COMMENTS

Reviewer #1 (Remarks to the Author):

All my concerns have been properly addressed.

Reviewer #2 (Remarks to the Author):

Compared to the original version, the authors have provided a careful revision that addressed all suggestions of the reviewers adequately. For that purpose, additional experiments were conducted that are convincing. Two minor comments are:

- Figure S14b: The stronger hydrophobic interactions are almost impossible to see from the figure.
- The language of the newly added text can be polished. This can be done during the editing of the manuscript after acceptance.

The principle strength and weaknesses of the manuscript outlined in my first review prevail, but I have no further suggestions for improvement.

Reviewer #3 (Remarks to the Author):

I think the authors have addressed the points raised in a satisfactory way, and I approve publication. However, I still think all biophysical data should be reported with two significant numbers, rather than in nM integers, which does not make sense. But I leave it up to the authors or editors to decide this.

Responses to Reviewer 1:

Comments: All my concerns have been properly addressed.

Response: We thank the reviewer for the positive comment.

Responses to Reviewer 2:

Comments: Compared to the original version, the authors have provided a careful revision that addressed all suggestions of the reviewers adequately. For that purpose, additional experiments were conducted that are convincing.

Response: We thank the reviewer for the positive comment.

Comments: Two minor comments are: - Figure S14b: The stronger hydrophobic interactions are almost impossible to see from the figure.

Response: As shown in Supplementary Fig. 14b, simnotrelvir forms hydrophobic interactions with seven S2 subsite residues including H41, C44, M49, Y54, M165, R188, and Q189. In comparison, the other three types of cyclic P2-segments only form hydrophobic interactions with three or four S2 subsite residues. To clearly show that the dithiazaspiro-proline group forms more hydrophobic interactions with the S2 subsite compared to other three

types of cyclic segments, we also presented the P2-residue interactions using the software Logplot in Supplementary Fig. 14c.

Comments: - The language of the newly added text can be polished. This can be done during the editing of the manuscript after acceptance.

Response: Following the reviewer's suggestion, a sentence-by-sentence editing of the newly added text has been done in the revised manuscript.

Comments: The principle strength and weaknesses of the manuscript outlined in my first review prevail, but I have no further suggestions for improvement.

Response: we thank the reviewer for the comment.

Responses to Reviewer 3:

Comments: I think the authors have addressed the points raised in a satisfactory way, and I approve publication.

Response: We are grateful for the positive comments given by the reviewer.

Comments: However, I still think all biophysical data should be reported with two significant numbers, rather than in nM integers, which does not make sense. But I leave it up to the authors or editors to decide this.

Response: Following the comment raised by Reviewer 2 in the first round of review, we used nM for the potency (IC50 and EC50) of all compounds, which makes it easier for the readers to compare the potency of different compounds. In order to be consistent with the biochemical data (potency), we also used nM integers for Kd values in Fig. 1&Table S1, the biophysical data in the manuscript.

With these changes we hope we have addressed all comments. We would like to thank the reviewers again for their constructive, professional and helpful suggestions, and the editor for handling this manuscript.